# An electrodiffusive neuron-extracellular-glia model for exploring the genesis of slow potentials in the brain

**Marte J. Sætra**[1], **Gaute T. Einevoll**[2,3,4], **Geir Halnes**[2,4]*

**1** Department of Numerical Analysis and Scientific Computing, Simula Research Laboratory, Oslo, Norway, **2** Centre for Integrative Neuroplasticity, University of Oslo, Oslo, Norway, **3** Department of Physics, University of Oslo, Oslo, Norway, **4** Department of Physics, Norwegian University of Life Sciences, Ås, Norway

* geir.halnes@nmbu.no

**Data Availability Statement:** All relevant data are within the manuscript. As stated in the manuscript, the model code can be downloaded from: https://github.com/CINPLA/edNEGmodel and https://github.com/CINPLA/edNEGmodel_analysis.

## Abstract

Within the computational neuroscience community, there has been a focus on simulating the electrical activity of neurons, while other components of brain tissue, such as glia cells and the extracellular space, are often neglected. Standard models of extracellular potentials are based on a combination of multicompartmental models describing neural electrodynamics and volume conductor theory. Such models cannot be used to simulate the slow components of extracellular potentials, which depend on ion concentration dynamics, and the effect that this has on extracellular diffusion potentials and glial buffering currents. We here present the electrodiffusive neuron-extracellular-glia (edNEG) model, which we believe is the first model to combine compartmental neuron modeling with an electrodiffusive framework for intra- and extracellular ion concentration dynamics in a local piece of neuro-glial brain tissue. The edNEG model (i) keeps track of all intraneuronal, intraglial, and extracellular ion concentrations and electrical potentials, (ii) accounts for action potentials and dendritic calcium spikes in neurons, (iii) contains a neuronal and glial homeostatic machinery that gives physiologically realistic ion concentration dynamics, (iv) accounts for electrodiffusive transmembrane, intracellular, and extracellular ionic movements, and (v) accounts for glial and neuronal swelling caused by osmotic transmembrane pressure gradients. The edNEG model accounts for the concentration-dependent effects on ECS potentials that the standard models neglect. Using the edNEG model, we analyze these effects by splitting the extracellular potential into three components: one due to neural sink/source configurations, one due to glial sink/source configurations, and one due to extracellular diffusive currents. Through a series of simulations, we analyze the roles played by the various components and how they interact in generating the total slow potential. We conclude that the three components are of comparable magnitude and that the stimulus conditions determine which of the components that dominate.

**Funding:** This work was funded by the Research Council of Norway (Norges Forskningsråd: https://www.forskningsradet.no) via the BIOTEK2021 Digital Life project 'DigiBrain', grant no 248828 (received by GTE), and EU project (Horizon 2020) HBP SGA3 Grant agreement ID: 945539 (received by GTE). The funders had no role in study design, data collection and analysis, decision to publish, or preparation of the manuscript.

**Competing interests:** The authors have declared that no competing interests exist.

## Author summary

A common experimental method for investigating brain activity is to measure the electric potential outside neurons. These recordings usually only capture the high-frequency part of the potential while ignoring frequencies below a set cut-off between 0.1 and 1 Hz. Therefore, standard recordings cannot tell us what the slow frequency potentials might say about on-going brain activity. Most computational models are also not suited in this regard. They ignore two possibly important contributions to the slow potentials, namely the diffusion of ions in the extracellular space and a cell type called astrocytes that contribute to keeping an appropriate chemical environment for neurons. To overcome this, we here present what we call the electrodiffusive neuron-extracellular-glia (edNEG) model for exploring the genesis of slow potentials in the brain. We use the model to study the contributions from neurons, astrocytes, and diffusive currents to the slow potentials and show that the model predicts that they are on the same order of magnitude.

## Introduction

A common experimental method for investigating brain activity is to measure the electric potential, either at the scalp (EEG), at the cortical surface (ECoG), or in the extracellular space inside the brain (LFP or spikes) [1]. These recordings are traditionally done using a low-frequency filter, with a cut-off frequency normally set somewhere between 0.1 and 1 Hz (see, e.g., [2–4]). Frequency components below this threshold are often referred to as slow potentials, standing potentials, sustained potentials, or DC potentials. We will here use the term slow potentials. The information that slow potentials might provide about on-going brain activity is discarded from standard recordings.

A multitude of brain processes have been associated with slow potentials, including both physiological phenomena, such as brain-state transitions and readiness potentials, and pathological phenomena, such as spreading depression, stroke, and epilepsy [4]. Slow potentials are often correlated with changes in extracellular ion concentrations, and especially with rises in the extracellular $K^+$ concentration. Such correlations are, for example, found regularly in studies of seizure activity [4, 5] and spreading depression [6, 7]. In layered brain regions such as the cortex and hippocampus, concentration shifts are inhomogeneous across layers, and during seizure activity and spreading depression, concentration gradients arise between deeper (cell-body) layers and superficial (dendritic) layers. Slow-potential shifts are normally reported to follow similar depth profiles as the extracellular $K^+$ concentration (see, e.g., [5, 8–13]).

There are three main candidate mechanisms (M1-M3) for explaining the correlation between gradients in ion concentrations and slow potentials, all of which were discussed in a recent review paper addressing how such potentials (there called DC potentials) arise during spreading depression [7]. These are (M1) neural sink/source configurations, (M2) glial sink/source configurations, and (M3) electrodiffusion (Fig 1). The two first (M1-M2) are conceptually similar: When the chemical environment varies with depth, cellular parts in superficial versus deeper layers will have different ionic reversal potentials over their membranes. Expectedly, this will lead to gradients in the membrane- and intracellular potentials, and thus to intracellular steady-state currents, even when the cells are otherwise inactive. Since currents always travel in closed loops, an intracellular current that, for example, goes towards deeper layers through (M1) neural dendrites or (M2) a glial syncytium, requires inward currents entering the cells (sinks) in the superficial layers, and outward return currents (sources) in the deeper layers. Such a sink and source configuration requires an extracellular current going towards

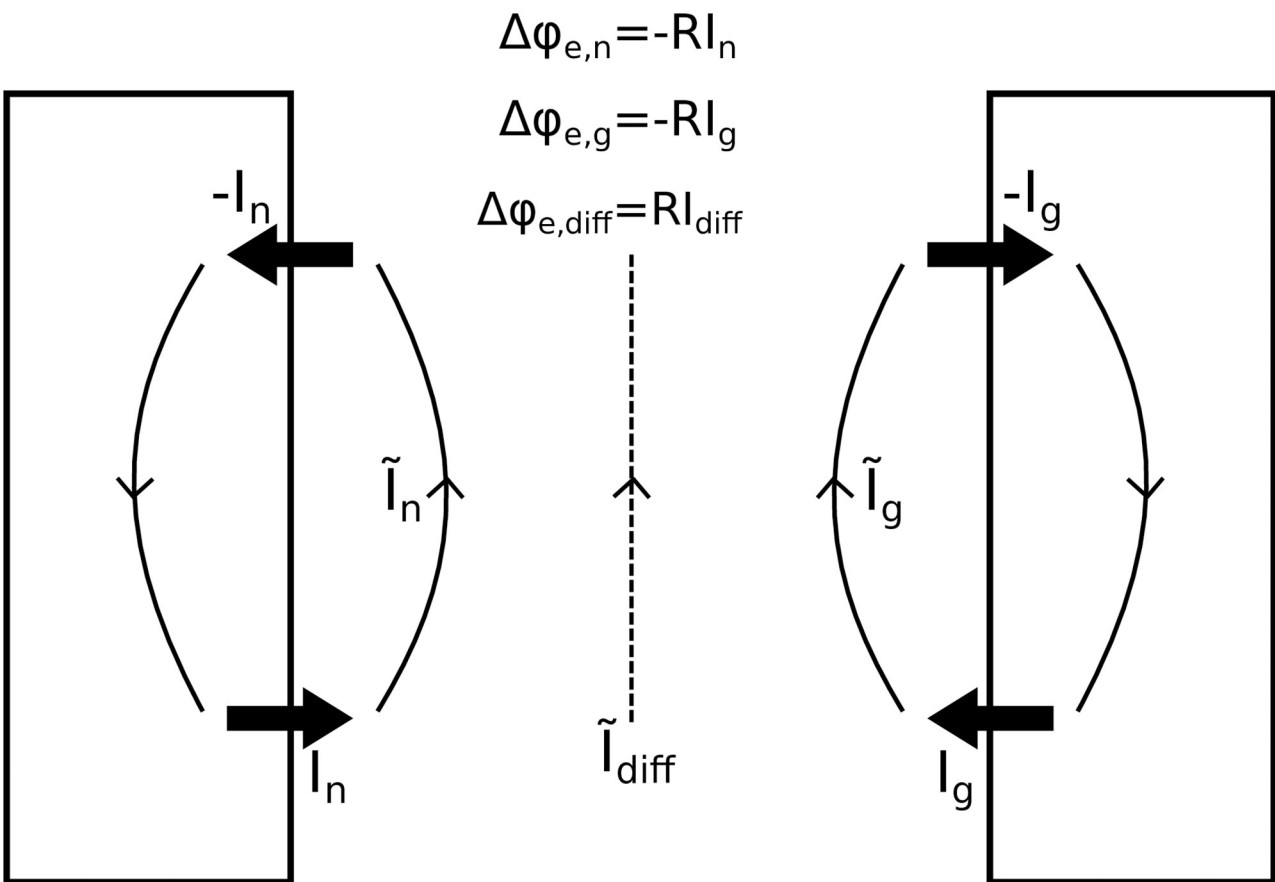

**Fig 1. Neuronal, glial, and diffusive contributions to the extracellular potential.** Neural sink/source configurations (left), glial sink/source configurations (right), and diffusion (middle) give contributions to the extracellular potential gradient $\Delta\phi_e$: $\Delta\phi_e = \Delta\phi_{e,n} + \Delta\phi_{e,g} + \Delta\phi_{e,\text{diff}}$. For a two-layered, one-dimensional system, these contributions can be computed as indicated in the top of the figure, where $R$ denotes the total extracellular resistance between the source and sink locations. Currents in the system are the neural membrane sources/sinks ($\pm I_n$), glial membrane sources/sinks ($\pm I_n$), extracellular diffusive currents ($\tilde{I}_{\text{diff}}$), and extracellular field currents due to the neuronal ($\tilde{I}_n$) and glial ($\tilde{I}_g$) source/sink configurations. The three contributions to $\Delta\phi_e$ are not independent. Current loop-completion requires that $I_n + I_g = \tilde{I}_n + \tilde{I}_g + \tilde{I}_{\text{diff}}$, indicating that $I_n \neq \tilde{I}_n$ and $I_g \neq \tilde{I}_g$. When $\Delta\phi_{e,n}$ and $\Delta\phi_{e,g}$ are computed from membrane current sources, as in standard volume conductor theory, the contribution from diffusion ($\Delta\phi_{e,\text{diff}}$) can be seen as a "correction" term, accounting for the fact that also diffusion contributes to completing the current loops. See the Analysis section in Methods for a full description of how we calculated $\Delta\phi_{e,n}$, $\Delta\phi_{e,g}$, and $\Delta\phi_{e,\text{diff}}$.

the superficial layers in order to complete the loop, and thus a gradient in the extracellular potential. The spatial $K^+$ buffering current in astrocytes [14] is a well-known example of such a slow current loop. In addition to M1 and M2, diffusion of ions along extracellular concentration gradients can (M3) give rise to a so-called diffusion current. The diffusion current is a direct function of the concentration differences and the diffusion constants of the involved ions, and will contribute to completing the current loops between the membrane sources and sinks (see, e.g., [15]). The three components (M1-M3) are therefore not independent (see Fig 1, figure caption).

Computational modeling in neuroscience has largely focused on simulating the fast electrical activity of neurons and networks of such, while ignoring other components of brain tissue, such as glia cells and the extracellular space. Within that paradigm, multicompartment neuron models are typically based on a combination of a Hodgkin-Huxley type formalism for membrane mechanisms (see, e.g., [16, 17]), and cable theory for how signals propagate in dendrites

and axons (see, e.g., [18, 19]). Two underlying assumptions in these standard models are that the neurodynamics is unaffected by changes in (i) extracellular potentials and (ii) extracellular ion concentrations. Models of this kind thus do not account for so-called ephaptic effects, where neurons may affect their neighbors non-synaptically through inducing changes in the extracellular environment [20]. Electric ephaptic effects have been the topic of many studies (see, e.g., [21–28]), as has the effect of changing ion concentrations on neurodynamics (see, e.g., [29–33]). The justification for neglecting such effects in standard simulations is that they often (and by assumption) are quite small, at least on the relatively short time-scale considered in most neural simulations.

Computational modeling of ECS potentials is typically done by combining the above mentioned standard models with volume conductor theory [34] through a forward modeling scheme consisting of two steps [3]: In step 1, the neurodynamics is computed under the assumption of constant extracellular conditions, and then, in step 2, the extracellular potentials are computed as a function of the transmembrane neural currents found in step 1. This modeling scheme may be suited to capture the fast dynamics of extracellular potentials, which are believed to be predominantly generated by neural activity and synaptic input to neurons [1, 3], i.e., by the first (M1) of the mechanisms discussed above. However, in the cases with large extracellular concentration gradients, it has been estimated that diffusion (M3) can give non-negligible contributions to extracellular potentials, even to frequency components on the order of ∼1 Hz, i.e., above the normal cut-off frequency [15, 35, 36]. Via effects on ionic reversal potentials, concentration shifts will also affect the other mechanisms (M1-M2). Hence, the standard framework is not applicable to studies of slow potentials, which depend on slower mechanisms involving concentration changes and their effects on extracellular diffusive currents, glial current sources, and neural current sources.

As they might involve both diffusion potentials and concentration-dependent effects on cellular dynamics, computational modeling of slow potentials on the spatial scale of tissue requires a self-consistent electrodiffusive framework that ensures conservation of ions and charge, and a physically consistent relationship between ion concentrations and electrical potentials in both the intra- and extracellular space [33]. Domain-type models that are consistent in this regard (see, e.g., [37–39]), have not accounted for the differential expression of membrane mechanisms in dendrites versus somata, or do not include a glial domain [33]. As cellular contributions to extracellular potentials depend on the spatial separation between transmembrane current sinks and sources [3], a model of slow potentials should, at least in some crude way, account for differences between somatic and dendritic current sources. Such differences are also the likely cause of the extracellular ion concentration gradients that occur under some conditions. For example, the concentration shifts seen during spreading depression have been suggested to originate in superficial layers of hippocampus and cortex, and to depend strongly on ion channel openings in the apical dendrites of pyramidal cells [6, 7, 10, 40–42].

Recently, we developed the electrodiffusive Pinsky-Rinzel (edPR) model, which we believe is the first model that combines a neural model with soma and dendrites with biophysically consistent modeling of ion concentrations, electrical charge, and electrical potentials in both the intra- and extracellular space [33]. In that work, we equipped the well-established two-compartment Pinsky-Rinzel model [43] with ion pumps, cotransporters, and equations for ion concentration dynamics in the intra- and extracellular space. The objective was to supply the neuroscience community with a model that can simulate neural dynamics not only under physiological conditions, where the homeostatic machinery succeeds in maintaining ion concentrations close to baseline, but also under pathological conditions, where homeostasis is incomplete, so that ion concentrations change over time.

Two potentially important contributors to slow potentials that were not accounted for in the edPR model were glia cells and effects of cellular swelling or shrinkage. In particular, a type of glia cells called astrocytes is known to be important for regulating the ionic content of the ECS [44], and especially for the uptake of excess $K^+$ that may develop during neuronal hyperactivity [14, 45–47]. Furthermore, when ion concentrations change in neurons, astrocytes, and the ECS, it will cause osmotic pressure gradients over the cellular membrane. This can lead to cellular swelling or shrinkage [48–51], which in turn will alter the ionic concentrations in the swollen or shrunken volumes. Cellular swelling and a corresponding shrinkage of the ECS is, for example, an important trademark of pathological conditions such as seizures and spreading depression [52–54].

To account for the main mechanisms behind slow potentials, we here expand the edPR model by including a glial domain and accounting for neuronal and glial swelling. We refer to the new model as the electrodiffusive neuron-extracellular-glia (edNEG) model and consider it to be a minimal model that includes the main machinery responsible for slow potential generations in a "unit" piece of brain tissue, i.e., a tissue volume of the size that on average will contain a single neuron, and the ECS and glial ion uptake that it has to its disposal. The edNEG model has six compartments, two for each of the three domains. It has the functionality that it (1) keeps track of all ion concentrations ($Na^+$, $K^+$, $Ca^{2+}$, and $Cl^-$) in all compartments, (2) keeps track of the electrical potential in all compartments, (3) has different ion channels in neuronal soma and dendrites so that the neuron can fire somatic action potentials (APs) and dendritic calcium spikes, (4) contains the neuronal and glial homeostatic machinery that maintains a realistic dynamics of the membrane potential and ion concentrations, (5) accounts for transmembrane, intracellular, and extracellular ionic movements due to both diffusion and electrical migration, and (6) accounts for cellular swelling of neurons and glia cells due to osmotic pressure gradients.

Using the edNEG model, we here simulate slow potentials occurring under conditions of varying (1) stimulus type (constant current injection versus synaptic), (2) stimulus location (somatic, dendritic, or uniformly distributed), and (3) stimulus strength (evoking physiological versus pathological activity), and study how the various mechanisms (M1-M3) defined above contribute to the total slow potential under the various conditions. As a general, qualitative insight, the edNEG model predicts that the three mechanisms give contributions to the slow potentials that are of the same order of magnitude, and that the mechanism that contributes the most differs between different stimulus conditions.

## Results

### An electrodiffusive tissue model with neuron-glia interactions

The architecture of the edNEG model is illustrated in Fig 2. It contains a neuronal, extracellular (ECS), and glial domain, all of which were modeled with two compartments. We use the term "domain" here, because the edNEG model is best interpreted as a tri-domain model, i.e., it represents the average neural, glial, and extracellular activity occurring in a "unit" piece of brain tissue, i.e., a tissue volume of the size that on average will contain a single neuron, and the ECS and glial ion uptake that it has at its disposal.

For the neuronal domain, the two compartments represent the neural membrane characteristics in somatic (bottom) and dendritic (top) layers. In the edNEG model, these compartments were connected with an intra-domain resistance similar to the intracellular resistance in the Pinsky-Rinzel model [33, 43], so that the neuronal domain was essentially modeled in the same way as a single two-compartment neuron model with "explicit" geometry. For the ECS, the two compartments represent the average ECS that the single neuron has at its disposal

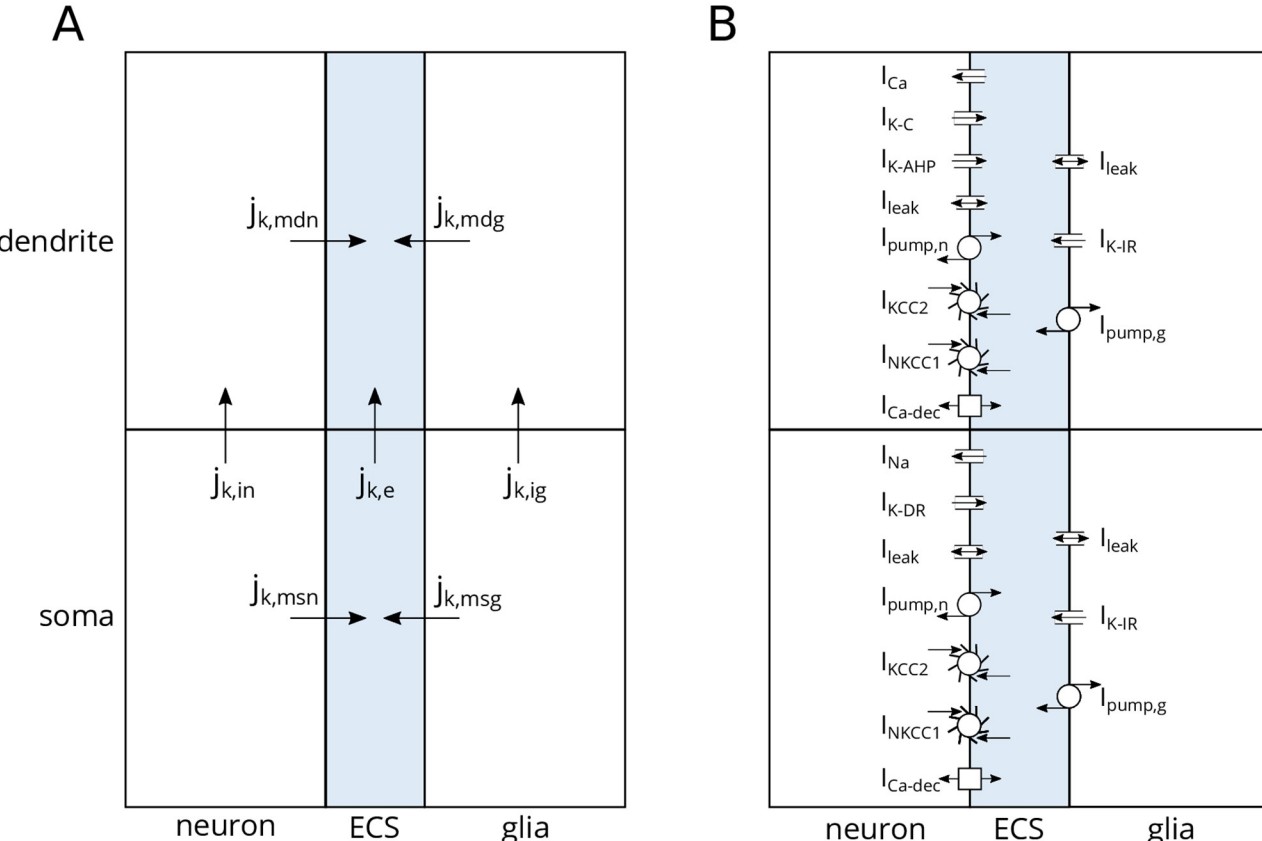

**Fig 2. Model architecture. (A)** The edNEG model contained three domains (neuron, index *n*, ECS, index *e*, and glia, index *g*). Initial neuronal/extracellular/glial volume fractions were 0.4/0.2/0.4. Each domain contained two compartments (soma level, index *s*, and dendrite level, index *d*). Ions of species *k* were carried by two types of fluxes: transmembrane (index *m*) fluxes ($j_{k,msn}$, $j_{k,mdn}$, $j_{k,msg}$, $j_{k,mdg}$) and intra-domain fluxes in the neuron ($j_{k,in}$), the ECS ($j_{k,e}$), and the glia cell ($j_{k,ig}$). An electrodiffusive framework was used to calculate ion concentrations and electrical potentials in all compartments. **(B)** The neuronal membrane contained the same mechanisms as in [33]. Both compartments contained Na$^+$, K$^+$, and Cl$^-$ leak currents ($I_{leak}$), 3Na$^+$/2K$^+$ pumps ($I_{pump,n}$), K$^+$/Cl$^-$ cotransporters ($I_{KCC2}$), Na$^+$/K$^+$/2Cl$^-$ cotransporters ($I_{NKCC1}$), and Ca$^{2+}$/2Na$^+$ exchangers ($I_{Ca-dec}$). The soma contained Na$^+$ and K$^+$ delayed rectifier currents ($I_{Na}$ and $I_{K-DR}$), and the dendrite contained a voltage-dependent Ca$^{2+}$ current ($I_{Ca}$), a voltage-dependent K$^+$ afterhyperpolarization current ($I_{K-AHP}$), and a Ca$^{2+}$-dependent K$^+$ current ($I_{K-C}$). The glial membrane mechanisms were taken from [47], and they were the same in both compartments. They included Na$^+$ and Cl$^-$ leak currents ($I_{leak}$), inward rectifying K$^+$ currents ($I_{K-IR}$), and 3Na$^+$/2K$^+$ pumps ($I_{pump,g}$). The edNEG model also accounted for cellular swelling due to osmotic pressure gradients across the membranes.

surrounding its somata (bottom) and dendrites (top). Finally, the glia cells most involved in ion homeostasis, the astrocytes, are typically interconnected via gap junctions into a continuous syncytium. The glial compartments could thus be interpreted not as two compartments of a single glia cell but rather as a representative for the average glial buffering surrounding the neural somata (bottom) and dendrites (top). To keep notation short, we will in the following refer to the neural, ECS, and glial domains, simply as the neuron, the ECS, and the glia cell.

The neuron and the ECS were adopted from the previously published electrodiffusive Pinsky-Rinzel (edPR) model [33] and modified slightly (see Methods). The glial model was based on a previous model for astrocytic spatial buffering [47] and added to the edPR model so that both the neuron and glia cell interacted with the ECS. Unlike the previous neuron [33] and glia [47] models that it was based upon, the edNEG model was constructed so that it also accounted for cellular swelling due to osmotic pressure gradients. We implemented the edNEG model using the electrodiffusive Kirchoff-Nernst-Planck framework (KNP) [33, 47], which consistently outputs the voltage- and ion concentration dynamics in all compartments.

Both the neuron and glial domain contained cell-specific and ion-specific passive leakage channels, cotransporters, and ion pumps (Fig 2) that ensured a stable ion balance in the system. The neuron contained additional active ion channels that were different in the somatic versus dendritic compartment, enabling it to fire somatic action potentials (AP) and dendritic $Ca^{2+}$ spikes. Both glial compartments contained inward rectifying $K^+$ channels. All included membrane mechanisms are summarized in Fig 2 and described in further detail in the Methods section.

## edNEG modeling of physiological and pathological activity

In standard (Hodgkin-Huxley type) neuron models, the key dynamical variable is the membrane potential. In addition to modeling the membrane potential, the edNEG model keeps track of the ion concentrations and electric potentials in all neuronal, glial, and extracellular compartments, as well as changes in compartment volumes due to osmotic gradients. It also accounts for the effect that changes in these variables may have on neuronal firing properties.

When the neuron is active, the exchange of ions due to AP firing will be counteracted by the stabilizing mechanisms striving to restore baseline concentration gradients. Hence, for moderately low neuronal firing, the edNEG model will enter a dynamic steady-state where homeostasis is successful, and firing can prevail for an arbitrarily long period of time without ion concentrations diverging far off from baseline. For a too-high neuronal activity level, the stabilizing mechanisms of the edNEG model will fail to keep up, and gradual changes in ion concentrations will lead to gradual changes in neuronal firing properties, and eventually to ceased AP firing. We will refer to these two classes of activity patterns as physiological and pathological activity, respectively.

Physiological (here used meaning normal/healthy) activity of the edNEG model is illustrated in Fig 3, which shows how the membrane potentials, the extracellular ion concentrations, and the volumes vary during a 1400 s simulation. The neuron received a stimulus from $t = 1\,s$ to $t = 600\,s$ that made it fire at 1 Hz. Fig 3A shows the neuronal and glial membrane potentials in the soma layer over the full simulation period, and Fig 3B illustrates the shape of an AP. Fig 3B also shows that the system had a stable resting state. Before stimulus onset, the neuron and glia cell rested at their resting potentials of approximately −67 mV and −84 mV, respectively.

In Fig 3C and 3D, we have plotted the extracellular ion concentrations of all ion species ($Na^+$, $K^+$, $Cl^-$, $Ca^{2+}$) in the soma and dendrite layer, respectively. Values are given in terms of their deviance from baseline values. As the soma contained no $Ca^{2+}$ channels, variations in the extracellular $Ca^{2+}$ concentrations were very small, although not strictly zero, since minor concentration shifts could occur due to electrodiffusion of $Ca^{2+}$ between the soma and dendrite layer. For the other ion species, as well as for $Ca^{2+}$ ions in the dendrite layer, we see that the lines appeared to have a certain thickness while the neuron was firing. The thickness arose because of changes in the ion concentrations on a short time scale. Had we zoomed in, we would see that the lines had a zig-zagging shape, most pronounced for the $K^+$ concentrations, where the upstroke would reflect the efflux of $K^+$ during the repolarization phase of an AP, while the downstroke would reflect the stabilizing mechanisms that were active between APs, working to restore the baseline concentrations.

The stabilizing recovery between APs was incomplete at the beginning of the simulation, and the ion concentrations zig-zagged away from baseline for each consecutive AP. However, the gradual divergence from baseline increased the stabilizing activity, which prevented ion concentrations from deviating too dramatically from baseline. After a period of regular firing, the system entered a dynamic steady-state where the zigs and the zags became equal in

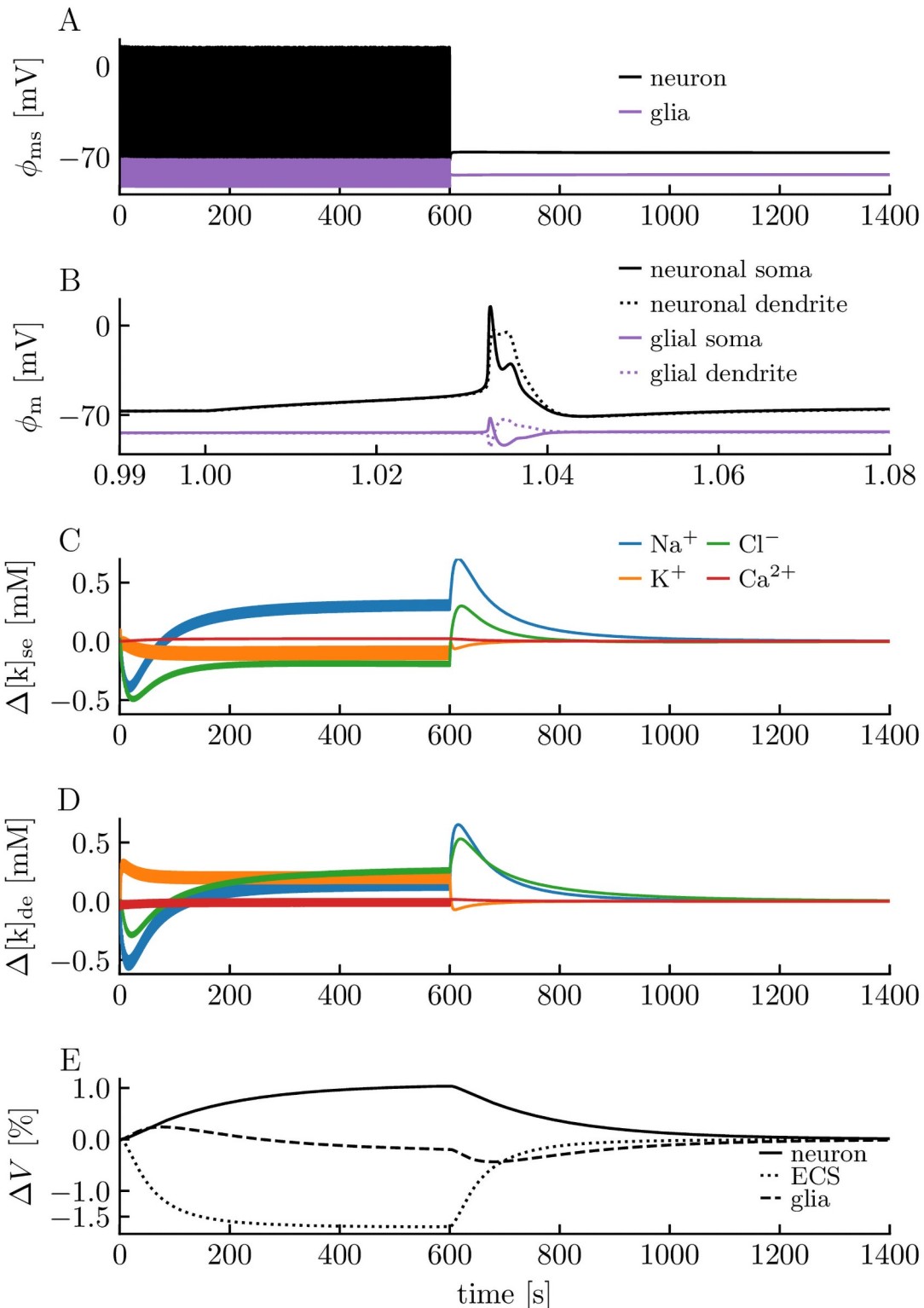

**Fig 3. edNEG modeling of physiological activity.** Model response to a 22 pA step-current injection to the somatic compartment of the neuron between $t = 1$ s and $t = 600$ s. The neuron responded with a firing rate of 1 Hz. The simulation covered 1400 s of biological time, and the last 800 s shows recovery to baseline. **(A)** The somatic membrane potential $\phi_{ms}$ of the neuron (black line) and the glia cell (purple line) for the full simulation period. **(B)** The membrane potential $\phi_m$ of the neuronal soma (black line), neuronal dendrite (black dotted line), glial "soma" (purple line), and glial "dendrite" (purple dotted line) for $t$

running from $t = 0.99$ s to $t = 1.08$ s. **(C)** Ion concentration dynamics of the extracellular space in the soma layer for all ion species ($Na^+$, $K^+$, $Cl^-$, $Ca^{2+}$) shown in terms of their deviance from baseline values. **(D)** Ion concentration dynamics of the extracellular space in the dendrite layer. **(E)** Volume dynamics of the three domains shown in terms of relative changes. Volume changes were computed as totals for the full domains (soma layer + dendrite layer). Initial neuronal/extracellular/glial volume fractions were 0.4/0.2/0.4.

magnitude, and the ion concentrations did not vary over time. In this simulation, $[K^+]_e$ deviated by maximally 0.4 mM, $[Na^+]_e$ by maximally −0.6 mM, $[Cl^-]_e$ by maximally −0.5 mM, and $[Ca^{2+}]_e$ by maximally −0.07 mM from the baseline ion concentrations during neuronal firing, which were not enough to have a visible impact on the regular firing of the neuron. Hence, the neuron in the edNEG model could fire regularly and continuously without dissipating its concentration gradients.

When the stimulus was turned off, the membrane potentials returned rapidly to values very close to the resting potential (Fig 3A), while the ion concentrations made a small "jump" before they returned more slowly towards baseline (Fig 3C and 3D). At the end of the simulation, ion concentrations had recovered the baseline values. If we define recovery (rather arbitrary) as the time it took for all ion concentrations to return to values less than 0.01 mM away from their resting baseline values, recovery took about 700 s, i.e., it occurred at about $t = 1300$ s. The fact that the membrane potentials were almost constant during the recovery of the reversal potentials indicates that the ion concentration recovery was due to a close-to electroneutral exchange of ions over the neuronal and glial membranes. Hence, the edNEG model predicts that "memories" of previous spiking history may linger in a neuron for several minutes, in the form of altered concentrations, even if it appears to have returned to baseline by judging from its membrane potential.

Pathological activity of the edNEG model, where the stabilizing mechanisms fail to keep up with the neuronal exchange, is illustrated in Fig 4. There, the neuron received a strong input current (150 pA) for seven seconds, giving it a high firing rate (Fig 4A and 4B). While the neuron fired, ion concentrations gradually changed (Fig 4C and 4D), causing a gradual depolarization of the neuron, which made it fire even faster. The neuron could tolerate this strong input for only a little more than 5 s before it became unable to re-polarize to levels below the AP firing threshold, and the firing ceased due to a permanent inactivation of the AP generating $Na^+$ channels. This condition, when a neuron is depolarized to voltage levels making it incapable of eliciting further APs, is known as depolarization block. It is a well-studied phenomenon, often caused by high extracellular $K^+$ concentrations [55]. We note that there are two main ways in which a neuron can be driven into depolarization block. The perhaps most well-studied scenario is that when the neuron fires so fast that it does not have time to repolarize properly between two consecutive action potentials. The action potential amplitude will then gradually decrease and tend to zero, repolarization will eventually fail, and the membrane potential will approach some depolarized equilibrium value. Neurons often have mechanisms (such as $Ca^{2+}$ channels coupled to afterhyperpolarizing $K^+$ channels) that limit firing rates and prevent them from entering firing-rate dependent depolarization block. An alternative scenario is when the depolarization block is caused by an increase in the extracellular $K^+$ concentration to values high above baseline due to, e.g., intense firing or impairment of the $Na^+/K^+$ pump [55]. An enhanced extracellular $K^+$ concentration will (i) depolarize the neuron due to an increased $K^+$ reversal potential, and at the same time, (ii) make repolarizing $K^+$ currents weaker due to the reduced $K^+$ gradient. Jointly, these two effects can put the neuron in a state where voltage-activated ion channels remain open, and the neuron gradually dissipates its concentration gradients approaching a depolarized equilibrium where it is unable to fire. The first type of depolarization block can be explored using models that do not model ion concentrations

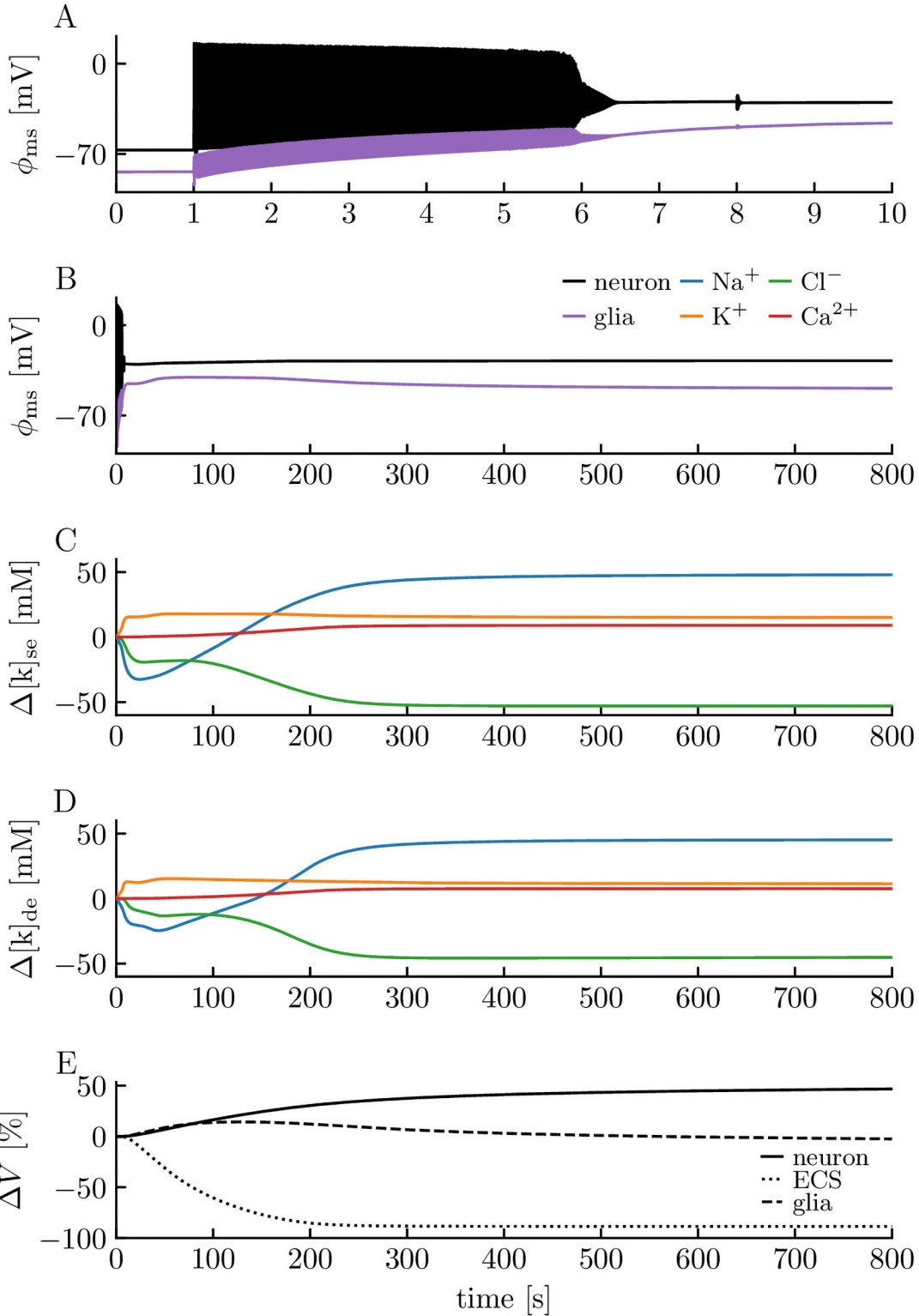

**Fig 4. edNEG modeling of pathological activity.** Model response to a 150 pA step-current injection to the somatic compartment of the neuron between $t = 1$ s and $t = 8$ s. **(A)** The neuron responded with an initial firing rate of 57 Hz, but both the firing rate and spike shapes varied throughout the simulation due to variations in the ion concentrations. Both the neuron (black line) and glia cell (purple line) experienced a gradual depolarization throughout the simulation, and the neuron eventually went into depolarization block, where it stayed throughout the rest of the simulation **(B)**. The gradually changing

dynamics patterns were due to activity-induced changes in ion concentrations (**C-D**). (**E**) The system experienced massive neuronal and glial swelling. Volume changes were computed as totals for the full domains (soma layer + dendrite layer).

explicitly (see, e.g., [56, 57], while the second type can only be studied in models that include ion concentration dynamics [31, 33] and is the one that we observe here.

When pathological activity had been induced, the system never returned to its baseline resting state, and the neuron never regained its ability to elicit APs. This has previously been referred to as a wave-of-death-like dynamics [58, 59]. It also resembles the neural dynamics seen under the onset of spreading depression [59], but during spreading depression, neurons tend to recover baseline activity after about one minute as the spreading depression wave passes [53]. Putatively, this recovery depends on $K^+$ being transported away from the local region by ECS electrodiffusion and spatial buffering through the astrocytic network, and quite likely also vascular clearance. As the edNEG model studied here represented a local and closed system, such spatial riddance of $K^+$ did not occur, but we anticipate that recovery might be observed if the edNEG model was expanded to a spatially continuous model or if $K^+$ was allowed to "leak" out of the system, relaxing towards its concentration in an external bath solution.

When the ion concentrations changed, so did the osmotic pressure gradients. This caused the neuron and glia cell to swell under both physiological (Fig 3E) and pathological activity (Fig 4E). The swelling was not dramatic during physiological firing, where the volume changes of the different domains were in the order of $\sim 1\%$. After the stimulus was turned off, the three domains recovered their original volume fractions. During pathological activity, the neuron lingered in depolarization block and continued to dissipate its concentration gradients so that cellular swelling went on for a long time. At the end of the simulation, ion concentrations were several tens of millimolar away from their baseline values, the neuron had swollen by 46.7%, and the glia cell and ECS had shrunken by 2.44% and 88.5%, respectively. In comparison, ECS shrinkage during spreading depression range from 40% to 78% [6, 60–64].

Having presented the edNEG model and its firing properties, we from here on shift our focus towards its prediction of extracellular slow potentials and how the various mechanisms (M1-M3) contribute to them. We note that in the edNEG model, changes in ion concentrations and volumes were modeled in a consistent manner, i.e., concentrations were computed as number of ions per volume, and hence, effects of volume dynamics were accounted for in the concentration dynamics. Apart from this, we did not directly explore the relationship between volume fractions and extracellular potentials.

### Extracellular slow potentials

In the edNEG model, ion concentrations and electric potentials were computed self-consistently in all compartments. As we took the ECS compartment in the dendrite layer as the reference point for the electric potential ($\phi_{de} = 0$), we will base our investigation of the ECS potential on the ECS potential in the soma layer ($\phi_{se}$).

When the neuron elicited an AP, a clear AP signature could be seen in the ECS potential in the soma layer (Fig 5A, dashed line). The characteristic extracellular spike consisted of a voltage drop (to about $-20$ mV) followed by a voltage increase (to about 20 mV). This biphasic response in the extracellular spike is well understood and has been examined in previous computational studies [21, 65, 66].

In the Methods section, we show that we can split the extracellular potential into three components that together sum up to the total potential. As we indicated in Fig 1, these are:

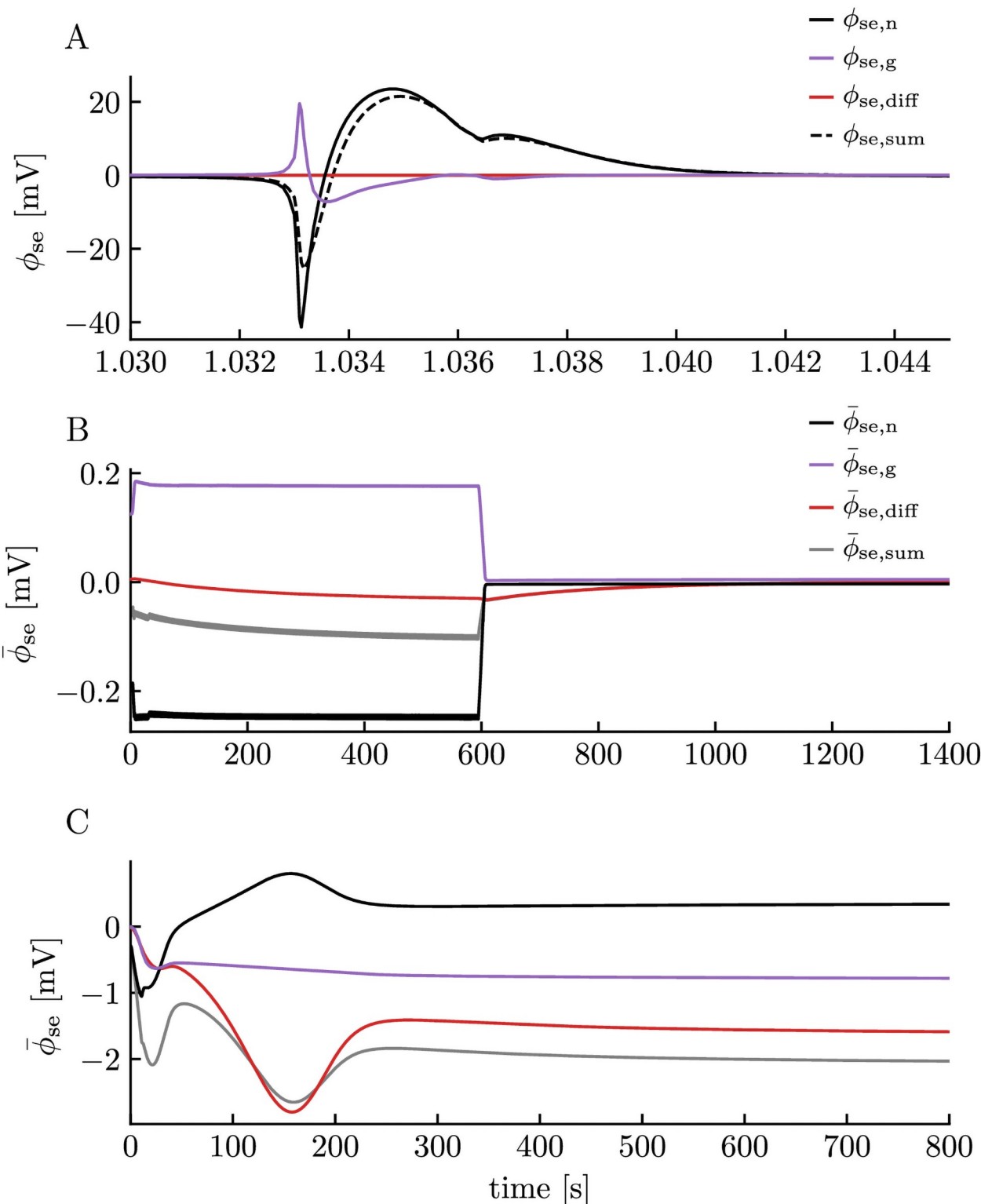

**Fig 5. Neuronal, glial, and diffusive components of the extracellular potential. (A)** The extracellular potential, split into components explained by standard volume conductor (VC) theory and neuronal currents ($\phi_{se,n}$), VC theory and glial currents ($\phi_{se,g}$), and a "correction" term explained by diffusive currents ($\phi_{se,diff}$). The sum of the three components ($\phi_{se,sum}$) is equal to the extracellular potential calculated in the edNEG model. See the Methods section for a description of how we calculated $\phi_{se,n}$, $\phi_{se,g}$, and $\phi_{se,diff}$. The simulation was the same as in Fig 3. **(B)** The neuronal ($\overline{\phi}_{se,n}$), glial

($\overline{\phi}_{se,g}$), and diffusive ($\overline{\phi}_{se,diff}$) components of the extracellular slow potential ($\overline{\phi}_{se,sum}$) from Fig 3, defined as the moving averages of $\phi_{se,n}$, $\phi_{se,g}$, $\phi_{se,diff}$, and $\phi_{se,sum}$ using a time window of 10 s. **(C)** The neuronal, glial, diffusive, and total component of the extracellular slow potential from Fig 4.

- $\phi_{se,n}$: Potential (in soma-layer) as predicted from the neural current sink/source configuration, using volume conductor theory.

- $\phi_{se,g}$: Potential (in soma-layer) as predicted from the glial current sink/source configuration, using volume conductor theory.

- $\phi_{se,diff}$: Potential (in soma-layer) as predicted from extracellular diffusion, using the KNP framework.

For readers familiar with the use of (standard) volume conductor (VC) theory for computing ECS potentials, the diffusion potential might be a new acquaintance. To give an intuitive understanding of how it affects the total potential, we start by noting that the sum of the two cellular components $\phi_{se,VC} = \phi_{se,n} + \phi_{se,g}$ will be the ECS potential as predicted from standard VC theory. Volume conductor theory is based on the principle of Ohmic current conservation, so that $\phi_{se,VC}$ is the potential needed to complete all current loops under the assumption that all extracellular currents are purely Ohmic, i.e., linearly dependent on the voltage gradient. However, if additional, diffusive currents are present, these will also contribute to current-loop completion, so that the real $\phi_{se} = \phi_{se,VC} + \phi_{se,diff}$ will deviate from $\phi_{se,VC}$. The diffusive component can thus be seen as a correction of the potential predicted from VC theory.

As expected, the extracellular spike signature (black, dashed line in Fig 5) is dominated by $\phi_{se,n}$ predicted from neuronal current sources and sinks (black, solid line). However, we also see that glial current sources and sinks contribute, especially during the initial negative peak. The glial contribution can be understood as follows: The initial peak is due to an inward current into the neuron in the soma-layer (depolarizing it), which requires a net current entering the extracellular compartment in the soma layer. This current comes partly from an extracellular current (from the dendritic to the somatic layer) and partly from an outward glial current. The latter gives rise to a glial "spike" (purple, solid line) with opposite polarity from the neuronal spike, and can be regarded as an ephaptic effect from the neuronal domain on the glial domain. As very small concentration changes occur on the short time-scale of AP firing, the diffusive component gives negligible contributions to the dynamics seen in Fig 5.

We note that the edNEG model dramatically overestimates the brief ECS voltage deflection seen during an AP. In experimental recordings, amplitudes in $\phi_e$ fluctuations are typically on the order of 0.1 mV [65], which is much smaller than that predicted by the edPR model. The discrepancy is an artifact mainly caused by the 1D approximation and the closed boundary conditions used in the edNEG model, which confine currents that, in reality, are 3D currents to go in only one spatial direction and to stay within the local system. Limiting the degrees of freedom in this manner essentially amounts to increasing the effective ECS resistance dramatically, thus causing larger voltage deflections. Importantly, while the edNEG model overestimates the amplitude of fast voltage deflections, it is reasonable to expect that it will give sound predictions of the slower components of extracellular potentials. The argument for this is that the closed boundary conditions applied in the simulations are equivalent to assuming periodic boundary conditions in the horizontal direction. Fig 5 thus simulates the extracellular spike in the hypothetical case of a population of perfectly synchronized neurons, i.e., one where all neurons in some brain region are identical and fire an AP at the exact same time. While such synchrony is unreasonable on the short time scale of AP firing, it seems like a reasonable assumption for processes on a longer time scale (arguments for this were also given in [15]).

Synchrony in slow signal components of, say, 0.1 Hz, simply means that all the neurons have the same average activity over a 10 s period.

From here on, we turn the focus away from the fast components of the ECS potential and focus on the slow potential. As a proxy of the slow potential, we used the average potential taken over a 10 s sliding window of simulated time. The slow potential in the ECS during the physiological and pathological neural activity are shown in Fig 5B (same simulation as in Fig 3) and Fig 5C (same simulation as in Fig 4), respectively. To keep notation short, we refer to contributions to the slow potential from neuronal sinks/sources (black curves), glial sinks/sources (purple curves), and diffusion (red curves), as the neuronal, glial, and diffusive $\overline{\phi}_{se}$-contributions, respectively.

During physiological conditions (Fig 5B), the neuronal $\overline{\phi}_{se}$-contribution (black curve) reflected the step current injection to the neural soma. This was not surprising, since the stimulus current dominated the neuronal membrane currents in the soma-layer during firing (S1A Fig). Since the step current injection amounted to a current sink, the neuronal $\overline{\phi}_{se}$-contribution was negative. The return current in the dendrite-layer was dominated by the $K^+$ afterhyperpolarization current (S1B Fig).

The glial $\overline{\phi}_{se}$-contribution (purple curve) was also step-like, but, similar to what we saw for the AP in Fig 5A, had the opposite polarity from the neuronal $\overline{\phi}_{se}$-contribution (see S1C and S1D Fig for the various current components contributing to glial sinks/sources). The glial $\overline{\phi}_{se}$-contribution was smaller than the neuronal contribution, so that the total slow potential (grey curve) was negative. The diffusive $\overline{\phi}_{se}$-contribution (red curve) was smaller in magnitude than the two other, but varied throughout the simulation as the extracellular ion concentration gradients changed. The concentration gradients remained in the system for a few hundred seconds after the stimulus had been turned off, and during this phase, the diffusive $\overline{\phi}_{se}$-contribution dominated the slow potential (grey and red lines coincide).

During pathological conditions, the neuron was stimulated so that it entered depolarization block and seized its AP activity after about 5 s of firing. The slow potential seen in Fig 5C was therefore mainly a result of gradual changes in ionic concentrations (cf. Fig 4C and 4D) and the effects that these had on neuronal and glial reversal potentials and extracellular diffusion. Towards the end of the simulation, the system reached a steady state where all concentrations and potentials remained at a constant value. Interestingly, in this final steady-state, the neuronal ($\sim 0.3$ mV), glial ($\sim -0.8$ mV), and diffusive ($\sim -1.5$ mV) $\overline{\phi}_{se}$-components added up to a total slow potential of about $-2$ mV. The steady-state is thus due to a balance between slow neuronal and glial current loops and a constant diffusion potential due to constant concentration gradients between the somatic and dendritic layers. The magnitude of the different membrane currents contributing to these current loops are illustrated in S2 Fig.

## Dependence of extracellular slow potentials on stimulus conditions

Next, we wanted to explore how the slow potentials generated in the edNEG model depend on stimulus conditions. To do this, we varied the stimulus strength, the stimulus type (constant injection versus synaptic), the stimulus position (soma, dendrite, or both), and the ion species that mediated the stimulus.

As we wanted to summarize the results from a number of simulations, we needed to simplify the analysis by selecting one slow potential measure per simulation. For physiological activity, simulations were instead run for 60 s, and the selected slow potential was defined as the average ECS potential taken over the last 10 s of the simulation, where the neuron fired with an (approximately) constant steady-state rate. For pathological conditions, simulations

were then run for 600 s. The neuron had then been in depolarization block for a long time, and all variables had settled on approximately constant values. Also in this scenario, the selected slow potential was computed as the average ECS potential taken over the last 10 s of the simulation.

In the first series of simulations, we evoked neuronal firing by applying constant current injections to the neuron for 60 s (Fig 6). As we have seen in earlier simulations, the magnitude of the injected current determined whether the neuron responded by settling on physiological steady-state firing (as in Fig 6A), or responded pathologically by entering depolarization block (as in Fig 6B). Fig 6C–6K summarizes the selected slow potentials from a number of simulations, where we have varied the stimulus conditions. As before, we decomposed the total slow potential (dashed curves) into the neuronal (black curves), glial (purple curves), and diffusive (red curves) $\overline{\phi}_{se}$-contributions.

Each of the panels in Fig 6C–6K summarizes a series of simulations, differing in terms of stimulus strength. The two different regimes, i.e., physiological versus pathological, are indicated by the curves splitting. The panels within each row differ in terms of where the stimulus was delivered, i.e., if it was delivered to the soma, the dendrite, or equally distributed over both compartments. The panels within a column differ in terms of which ion species that carried the stimulus current, i.e., whether it was a $K^+$ stimulus, a $Na^+$ stimulus, or a $Cl^-$ stimulus. We implemented the injection currents as inward positive currents across the neuronal membrane, affecting the intra- and extracellular ion concentrations in accordance with ion conservation (See Methods for details). The stimulus type and location are indicated above each panel in Fig 6C–6K.

The first thing that we notice is that, for pathological conditions (the rightmost half of the separated curves in each panel), the slow potential, and the neuronal, glial, and diffusive contributions to it, were more or less independent of stimulus strength, stimulus location, and stimulus type, i.e., they were the same in all panels. This indicates that the final pathological state is an equilibrium determined by the intrinsic dynamics of the system after the neuron had entered depolarization block, independently of what caused the neuron to enter depolarization block in the first place. In this state, the final slow potential was $\sim -2$ mV, and dominated by the diffusion potential $\sim -1.5$ mV, followed by the glial $\overline{\phi}_{se}$-contibution $\sim -0.8$ mV. The neuronal $\overline{\phi}_{se}$-contribution was smaller and positive $\sim 0.3$ mV, and thus acted to reduce the final slow potential.

For physiological conditions, we generally found that the slow potential (in the soma layer) was negative when the stimulus was delivered to the soma (Fig 6C, 6F and 6I), positive when the stimulus was delivered to the dendrite (Fig 6D, 6G and 6J), and close to (but not identical to) zero when the stimulus was distributed equally among the compartments (Fig 6E, 6H and 6K). Since the stimulus amounts to a current sink, it was not surprising that it biased the slow potential towards a negative value when applied to the soma layer and towards a positive value when applied to the dendrite layer, although the system contained other sources and sinks beside the stimulus. In all simulations, the absolute magnitude of the slow potential under physiological activity increased with stimulus strength.

Due to the stimulus bias, the polarity of the physiological slow potential always had the same polarity as the neuronal $\overline{\phi}_{se}$-contribution when stimulus was applied to only one compartment, and in general, the neuronal $\overline{\phi}_{se}$-contribution dominated slightly over the glial and diffusive $\overline{\phi}_{se}$-contributions. For example, with a $Na^+$ stimulus to the soma (Fig 6F), the diffusive and glial contribution canceled each other out, so that the slow potential was almost identical to the neural $\overline{\phi}_{se}$-contribution. In comparison, with a $K^+$ stimulus to the soma (Fig 6C), the diffusive $\overline{\phi}_{se}$-contribution was almost zero, and the slow potential ended up midways

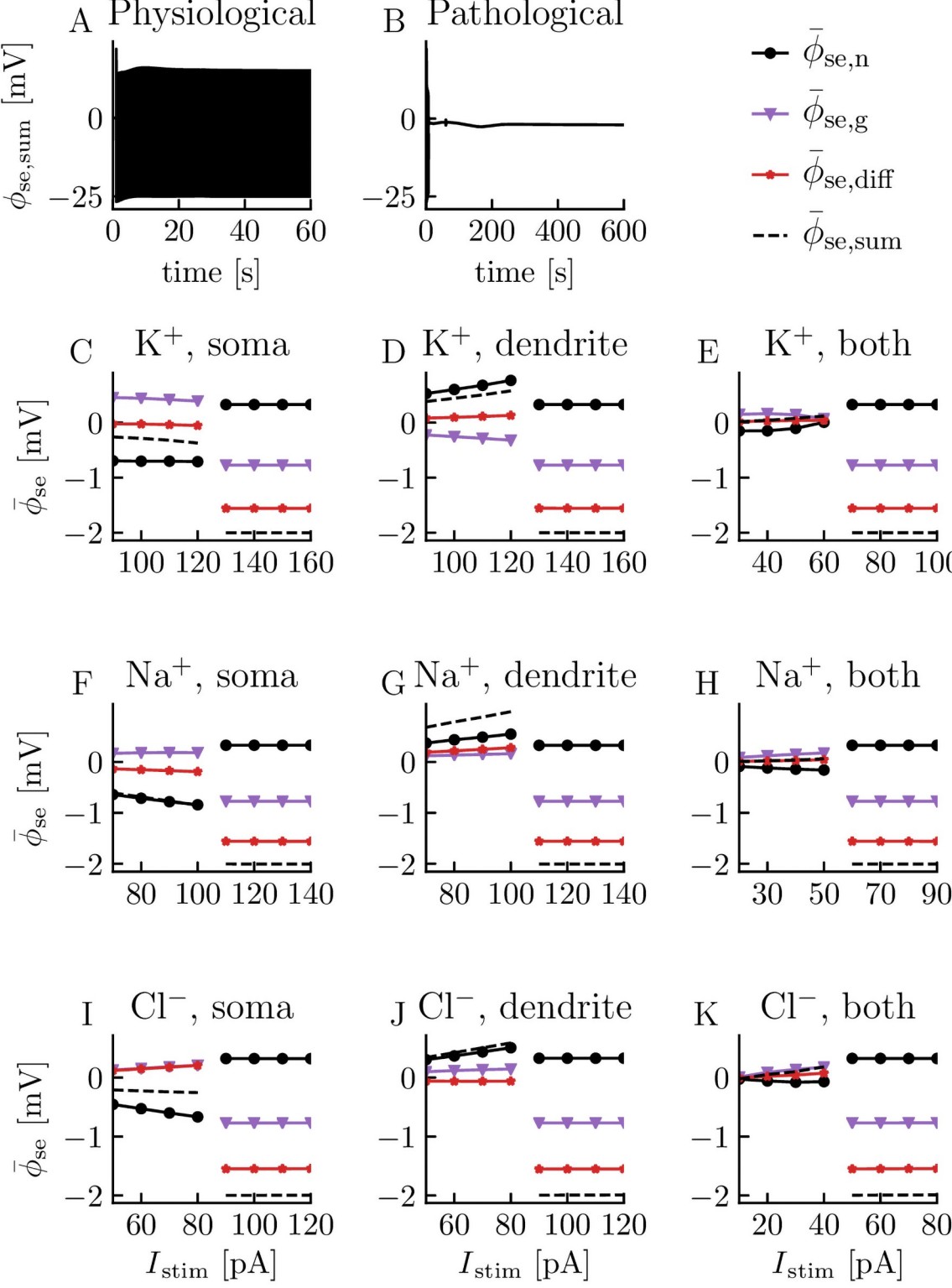

**Fig 6. Effects of neurons, glia, and diffusion on slow potentials under constant current injections.** (**A**) Extracellular potential during physiological steady-state firing caused by a 90 pA K$^+$ step current injection to the neuronal soma between $t = 1$ s and $t = 60$ s. Simulation ended at $t = 60$ s. (**B**) Extracellular potential during pathological conditions evoked by a 130 pA K$^+$ step current to the neuronal soma between $t = 1$ s and $t = 60$ s. The neuron entered depolarization block before the stimuli was turned off. Simulation ended at $t = 600$ s. (**C**)-(**K**) Extracellular slow potentials ($\overline{\phi}_{se,sum}$) as a function of the stimulus current $I_{stim}$. $\overline{\phi}_{se,sum}$ was split into

contributions from neuronal sinks/sources ($\overline{\phi}_{se,n}$), glial sinks/sources ($\overline{\phi}_{se,g}$), and diffusion ($\overline{\phi}_{se,diff}$). $\overline{\phi}_{se}$ was computed as the mean potential taken over the last 10 s of the simulations, i.e., from $t = 50$ s to $t = 60$ s for physiological cases and from $t = 590$ s to $t = 600$ s for pathological cases. The transition from physiological to pathological conditions corresponds to the curves breaking. The stimulus current was carried either by K$^+$, Na$^+$, or Cl$^-$ ions (different rows), and applied either to the somatic compartment, dendritic compartment, or both compartments of the neuron (different columns). The stimulus type and location are indicated above each panel.

between a positive glial $\overline{\phi}_{se}$-contribution and negative neuronal $\overline{\phi}_{se}$-contribution, where the latter was of greatest magnitude, so that the final slow potential became negative ($\sim -0.3$ mV). The largest (physiological) slow potential was seen when a Na$^+$ stimulus was delivered to the dendrite (Fig 6G). In that case, the neuronal, glial, and diffusive $\overline{\phi}_{se}$-contributions were all positive, adding up to a maximal slow potential of $\sim 1$ mV for a strong (but not pathologically strong) stimulus. Although the neuronal $\overline{\phi}_{se}$-contribution dominated, all the three contributions tended to be of the same order of magnitude.

To mimic more realistic stimulus conditions, we also stimulated the neuron with input from AMPA synapses (Fig 7). The synaptic current was composed of Na$^+$, K$^+$, and Ca$^{2+}$ ions in biologically realistic ratios (see the Synaptic current section in Methods). The synaptic input was delivered in terms of a Poissonian spike train, and the input frequency determined whether the response of the neuron was physiological (Fig 7B) or pathological (Fig 7C). The slow potentials selected from the two regimes were defined in the same way as for a constant current injection.

For synaptic stimuli, the pathological slow potentials (Fig 7) were close to identical to those obtained with constant current injections (Fig 6), which again indicates that the final state was an intrinsic equilibrium of the edNEG system. For stimuli in the physiological range, the slow potentials obtained with synaptic current (Fig 7D–7F) resembled those obtained with a constant Na$^+$ stimuli (Fig 6F–6H). This similarity is most likely a consequence of our AMPA current being dominated by Na$^+$ ions crossing the membrane (see Methods subsection titled Synaptic current), so that the AMPA and Na$^+$ stimuli gave rise to chemically similar effects in the system.

## Discussion

We presented the edNEG model for local ion concentration dynamics in a piece of brain tissue containing a neuronal, extracellular, and glial domain (Fig 2). The edNEG model was constructed with the aim to include the main categories of biophysical mechanisms involved in the generation of extracellular slow potentials. These included (i) a selection of neural and glial membrane mechanisms, including passive ion channels, ion pumps, and cotransporters in both cell types and additional active ion channels and synapses on the neuron, (ii) spatial heterogeneity, i.e., different ion channels in the soma and dendrites of neurons, and thus somatodendritic neural signaling, (iii) electrodiffusive ion concentration dynamics within all domains, inducing changes in ionic reversal potentials and diffusive currents evoking diffusion potentials, and (iv) neuronal and glial swelling due to concentration-dependent osmotic pressure gradients, and the effects that the corresponding volume changes had on local ion concentrations. We note that the spatial heterogeneity of the model restricts it to brain regions where neurons are organized in layers with all neurons having the same spatial orientation (such as, e.g., hippocampus).

Many previous models have modeled intra- and extracellular ion concentration dynamics, and have some of the same functionality as the edNEG model [29–32, 39, 58, 59, 67–97]. However, to our knowledge, the edNEG model is the first tri-domain type of electrodiffusive tissue

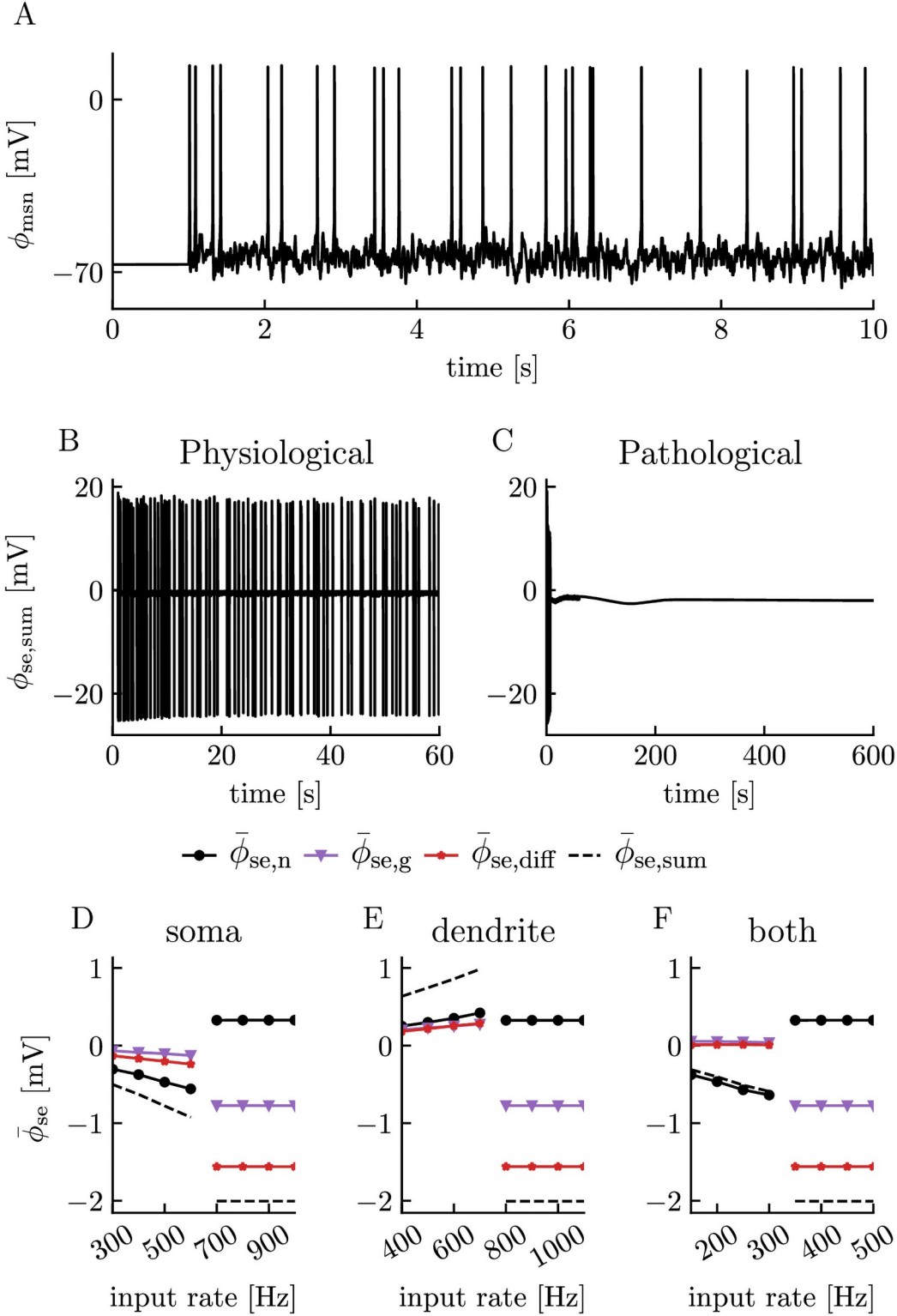

**Fig 7. Effects of neurons, glia, and diffusion on slow potentials under AMPA stimulus conditions.** (**A**) Somatic membrane potential of the neuron responding to a 300 Hz Poissonian AMPA spike train delivered to the soma between $t = 1$ s and $t = 10$ s. (**B**) Extracellular potential (oustide soma) during physiological steady-state firing obtained with a somatic synaptic input rate of 300 Hz between $t = 1$ s and $t = 60$ s. The simulation was ended at $t = 60$ s. (**C**) Extracellular potential (outside soma) during pathological conditions obtained with a synaptic input rate of 700 Hz between $t = 1$ s and

$t = 60$ s. The neuron entered depolarization block before the stimuli was turned off. Simulation was ended at $t = 600$ s. (**D**)-(**F**) Extracellular slow potentials ($\overline{\phi}_{e,sum}$) as a function of synaptic input rate. $\overline{\phi}_{se,sum}$ was split into contributions from neuronal sinks/sources ($\overline{\phi}_{se,n}$), glial sinks/sources ($\overline{\phi}_{se,g}$), and diffusion ($\overline{\phi}_{se,diff}$). $\overline{\phi}_{se}$ was computed as the mean potential taken over the last 10 s of the simulations, i.e., from $t = 50$ s to $t = 60$ s for physiological cases and from $t = 590$ s to $t = 600$ s for pathological cases. The transition from physiological to pathological conditions corresponds to the curves breaking. Synapses where located in (**D**) the somatic compartment, (**E**) dendritic compartment, or (**F**) both compartments of the neuron.

model that is spatially explicit in the vertical (soma → dendrite) direction, and therefore the first framework that can be used to simulate phenomena involving concentration gradients and slow potential gradients across layers in a self-consistent manner.

The main predictions of the edNEG model (Figs 6 and 7) were the following. Firstly, the $\overline{\phi}_{se}$-contributions from neurons, glia cells, and diffusion were of comparable magnitude, so that neither of them, as a generality, can be neglected. Secondly, for physiological neural firing, the slow potential was dominated by the neuronal activity, was biased to be negative at the location where the neuron received its stimulus (since the stimulus was a current-sink from the ECS and into the neuron), and increased with stimulus strength. Thirdly, for pathological conditions with the neuron entering depolarization block, the slow potential was eventually dominated by the intrinsic dynamics of the system. In this scenario, the slow potential was dominated by diffusion and glial buffering currents.

The predictions made with the edNEG model are bound to depend on a set of model choices, some of which were quite arbitrary. The general insights gained from the slow potential predictions are therefore at best qualitative: neurons, glia cells, and diffusion give comparable contributions to slow potentials, and their relative roles in generating them depend on stimulus conditions.

For the purpose of further discussion, we can divide the modeling choices into two classes. The first class concerns the choice of membrane mechanisms included in the cell models. In the edNEG model, we adopted a set of membrane mechanisms from previous models and did not perform any tuning in order to match the models to data obtained from a specific biological system. The choice of membrane mechanisms was somewhat arbitrary, and had we chosen another set of membrane mechanisms, we would likely arrive at somewhat different conclusions. As such, we believe that the edNEG model is valuable, not predominantly as "a final model", but rather through representing a framework for exploring how various ion channels in neuronal and glial membranes will affect slow potentials, both directly, through their contributions as current sinks or sources, or indirectly, through the concentration changes they may induce in the ECS. By modifying or exchanging the membrane mechanisms, the edNEG framework could, in principle, be re-tuned to represent other cell types (for frameworks for model-tuning, see, e.g., [98–100]), including other types of ion channels. Many ion channels, such as non-specific hyperpolarization-activated cation channels ($I_h$) and A-type K$^+$ channels ($I_A$) often have gradient distributions in neural dendrites [101], and to explore effects of this, it might be necessary to expand the edNEG model by including additional compartments for each of the domains. Such an expansion would require a re-writing of the model code (see the Numerical implementation section in Methods for source code), but would not require any fundamental changes of the conceptual framework that it was based upon, and should therefore not be too challenging for a user with training in programming.

The second class of modeling choices concerns the geometrical specifications of the system. Several electrodiffusive models have been developed for systems with explicit geometries (see e.g., [32, 102–108]). These models have the advantages that they are conceptually easy to interpret, i.e., the various compartments (often finite elements) exist side by side in a spatially

explicit meaning. However, they have the disadvantage that they are very computationally demanding, and applications have so far been limited to processes taking place on rather small spatial and temporal scales. When making a leap in spatial scale, as we did when constructing the edNEG model, we had to struggle with the question of how one best capture desired properties of neurons, ECS, and glia cells within an abstract tri-domain framework. This is a far from trivial problem. Domain-type models are inspired by the bi-domain model [109], which has been used to describe cardiac tissue as a bi-phasic continuum consisting of an intracellular and extracellular domain [110, 111]. Similar, tri-domain models (including neurons, ECS, and glia cells) have been used to simulate brain tissue [37–39]. Domain-type models are coarse-grained, meaning that a set of intra- and extracellular variables (e.g., voltages and ion concentrations), and exchanges between the intra- and extracellular domains, are defined at each point in space. In this context, a "point in space" refers to a spatial average taken over a volume that is big enough to span all domains. Previous tri-domain models of brain tissue [37–39] have been spatially explicit in the lateral direction, e.g., they have modeled spatial variations along the cortical surface, but not in the vertical direction, e.g., all variables are averages taken over all cortical layers. In those models, there was no over-distance intracellular coupling within the neural domain, since the intracellular domain of neurons does not form a continuous space at the tissue level. Differing from this, the edNEG model is spatially explicit in the vertical direction, and thus had to include such an intracellular, spatial coupling inside the neuronal domain, capturing the somatodendritic signaling. It is, however, not immediately evident what this coupling should be, since one in a domain-representation should not interpret the two layers explicitly as two pieces of the same, single neuron, but rather the average somatic activity and dendritic activity in a local region of tissue. In the edNEG model, we nevertheless defined this coupling in a manner giving a somatodendritic coupling similar to that of the single-cell Pinsky-Rinzel model. This might result in an over-estimate of the coupling, except in the hypothetical scenario discussed earlier, where all neurons in the tissue fire in perfect synchrony.

Generally, reducing the morphological complexity of a cell to a small number of compartments is not straightforward, and requires that compromises are made regarding what properties should be preserved under the reduction. This can be challenging even in the comparably simple case when one only models the intracellular electric dynamics of the cell [112], but more so when the model also includes the ion concentration dynamics and the extracellular environment. The impact of a transmembrane current on the membrane potential is inversely proportional to the membrane surface area of the compartment, while its impact on the intracellular ion concentration dynamics is inversely proportional to the compartment volume, meaning that various variables will scale differently with choices of geometrical parameters. In the edNEG model, we preserved the experimentally reported volume fractions of neuronal, extracellular, and glial domains. Apart from that, we did not attempt to base geometrical parameters for volumes, membrane surface areas, or cross-section areas for intra-domain flows directly on experimental measures. Instead, we treated these as somewhat free parameters, and specified them to values that gave reasonable dynamics for ion concentrations and potentials in the various compartments (see Methods), and preserved the firing patterns of the original Pinsky-Rinzel model [33, 43].

Finally, as we have argued earlier, the sealed boundary conditions used in the edNEG simulations were equivalent to having periodic boundary conditions, and we thus simulated the hypothetical scenario of a tissue region consisting of identical and synchronized cells. We chose to use this scenario because it has a clear interpretation, and as we argued, it makes sense to assume such synchrony when studying slower signals such as slow potentials. An alternative and equally simple setup would be to use open boundaries, for example allowing

the extracellular domain to be coupled with a constant bath-solution. Such a setup was not tested in the current implementation of the edNEG model, but could perhaps be valuable if one wished to mimic conditions with localized input to a neuron in an otherwise silenced region, and could putatively be used to model various slice experiments.

## Methods

### The Kirchoff-Nernst-Planck (KNP) framework for a three-times-two compartment model

We previously developed the electrodiffusive Pinsky-Rinzel model for a closed-boundary system containing $2 \times 2$ compartments, representing a soma, a dendrite, and extracellular space (ECS) outside the soma and dendrite [33]. The edNEG model expands the previous model by including an additional, glial domain, and accounting for osmotically induced volume changes. The three domains (neuron + ECS + glia) all consisted of two compartments, representing the soma layer and the dendrite layer. Within each layer, the neuron and glial domain interacted with the ECS through transmembrane currents (Fig 2). Volume changes were due to osmotic pressure gradients computed as functions of the ionic concentrations (see section titled Volume dynamics). Geometrical parameters, including initial volumes, are listed in Table 1. Dynamics of ion concentrations and electric potentials in all compartments were computed using the KNP framework [15, 32, 33, 47, 95].

**Electrodiffusion.** Two kinds of fluxes transport ions in the system: transmembrane fluxes and axial fluxes. The axial fluxes are driven by electrodiffusion, described by the Nernst-Planck equation so that the intracellular flux density of the neuron for ion species $k$ is expressed as:

$$j_{k,\text{in}} = -\frac{D_k}{\lambda_i^2} \frac{\gamma_k([k]_{dn} - [k]_{sn})}{\Delta x} - \frac{D_k z_k F}{\lambda_i^2 RT} \overline{[k]}_n \frac{\phi_{dn} - \phi_{sn}}{\Delta x}. \tag{1}$$

In Eq 1, $D_k$ is the diffusion constant, $\gamma_k$ (= 1 for all ions except $Ca^{2+}$) is the fraction of mobile ions of species $k$, that is, ions that are not buffered or taken up by the endoplasmatic reticulum, $\lambda_i$ is the tortuosity, which represents hindrances in free diffusion due to obstacles, $\gamma_k([k]_{dn} - [k]_{sn})/\Delta x$ is the longitudinal concentration gradient, $z_k$ is the charge number of ion species $k$, $F$ is the Faraday constant, $R$ is the gas constant, $T$ is the absolute temperature, $\overline{[k]}_n$ is the average intra-domain concentration, that is, $\gamma_k([k]_{dn} + [k]_{sn})/2$, and $(\phi_{dn} - \phi_{sn})/\Delta x$ is

**Table 1. Geometrical parameters.**

| Parameter | Value | Reference |
|---|---|---|
| $\Delta x$ (distance between the two layers) | $667 \cdot 10^{-6}$ m | [33] |
| $\alpha$ (intracellular coupling strength) | 2 | [33] |
| $A_m$ (membrane area of each cellular compartment) | $616 \cdot 10^{-12}$ m$^2$ | [33] |
| $A_i$ (intracellular cross-section areas) | $\alpha \cdot A_m$ | [33] |
| $A_e$ (extracellular cross-section area) | $308 \cdot 10^{-13}$ m$^2$ | |
| $V_{sn,0}$, $V_{dn,0}$ (initial neuronal volumes) | $1437 \cdot 10^{-18}$ m$^3$ | [33] |
| $V_{se,0}$, $V_{de,0}$ (initial extracellular volumes) | $718.5 \cdot 10^{-18}$ m$^3$ | [33] |
| $V_{sg,0}$, $V_{dg,0}$ (initial glial volumes) | $1437 \cdot 10^{-18}$ m$^3$ | |

We assumed a region thickness of 1.3 $\mu$m and consequently a $\Delta x$ that was half of this. The cellular compartment volumes and membrane areas correspond to spheres with radius 7 $\mu$m. The initial neuronal/extracellular/glial volume fractions were 0.4/0.2/0.4 [45].

the intra-domain potential gradient. Likewise, the extracellular flux densities and the glial intracellular flux densities are described, respectively, by

$$j_{k,e} = -\frac{D_k}{\lambda_e^2}\frac{[k]_{de} - [k]_{se}}{\Delta x} - \frac{D_k z_k F}{\lambda_e^2 RT}\overline{[k]}_e \frac{\phi_{de} - \phi_{se}}{\Delta x}, \tag{2}$$

$$j_{k,ig} = -\frac{D_k}{\lambda_i^2}\frac{[k]_{dg} - [k]_{sg}}{\Delta x} - \frac{D_k z_k F}{\lambda_i^2 RT}\overline{[k]}_g \frac{\phi_{dg} - \phi_{sg}}{\Delta x}. \tag{3}$$

All simulated ion species are mobile in the extracellular and glial space, thus, $\gamma_k$ is not included in Eqs 2 and 3. Diffusion constants, tortuosities, and intracellular fractions of mobile ions are listed in Table 2.

**Ion conservation.** To keep track of all ions in the system, we solve six differential equations for each ion species $k$. Conservation of ions gives:

$$\frac{dN_{k,sn}}{dt} = -j_{k,msn}A_m - j_{k,in}A_i, \tag{4}$$

$$\frac{dN_{k,se}}{dt} = +j_{k,msn}A_m - j_{k,e}A_e + j_{k,msg}A_m, \tag{5}$$

$$\frac{dN_{k,sg}}{dt} = -j_{k,msg}A_m - j_{k,ig}A_i, \tag{6}$$

$$\frac{dN_{k,dn}}{dt} = -j_{k,mdn}A_m + j_{k,in}A_i, \tag{7}$$

$$\frac{dN_{k,de}}{dt} = +j_{k,mdn}A_m + j_{k,e}A_e + j_{k,mdg}A_m, \tag{8}$$

$$\frac{dN_{k,dg}}{dt} = -j_{k,mdg}A_m + j_{k,ig}A_i, \tag{9}$$

where $N_k$ is the amount of substance, in units of mol. To find the change in $N_k$, all ion flux densities are multiplied by the area they go through. The variable $j_{k,m}$ represents the sum of all membrane flux densities of ion species $k$, and $j_{k,in}$, $j_{k,e}$, and $j_{k,ig}$ represent the axial flux densities. To find the ion concentrations [k], we divide the amounts of substance in a compartment

**Table 2. Diffusion constants, tortuosities, and intraneuronal fractions of mobile ions[†].**

| Parameter | Value | Reference |
|---|---|---|
| $D_{Na}$ (Na$^+$ diffusion constant) | $1.33 \cdot 10^{-9}$ m$^2$/s | [15] |
| $D_K$ (K$^+$ diffusion constant) | $1.96 \cdot 10^{-9}$ m$^2$/s | [15] |
| $D_{Cl}$ (Cl$^-$ diffusion constant) | $2.03 \cdot 10^{-9}$ m$^2$/s | [15] |
| $D_{Ca}$ (Ca$^{2+}$ diffusion constant) | $0.71 \cdot 10^{-9}$ m$^2$/s | [15] |
| $\lambda_i$ (intracellular tortuosity) | 3.2 | [47] |
| $\lambda_e$ (extracellular tortuosity) | 1.6 | [47] |
| $\gamma_{Na}, \gamma_K, \gamma_{Cl}$ (intraneuronal fractions of mobile ions) | 1 | [33] |
| $\gamma_{Ca}$ (intraneuronal fraction of mobile ions) | 0.01 | [33] |

[†] The table is adopted from [33].

by the compartment volume at the beginning of each time step:

$$[k]_{sn} = \frac{N_{k,sn}}{V_{sn}}, \tag{10}$$

$$[k]_{se} = \frac{N_{k,se}}{V_{se}}, \tag{11}$$

$$[k]_{sg} = \frac{N_{k,sg}}{V_{sg}}, \tag{12}$$

$$[k]_{dn} = \frac{N_{k,dn}}{V_{dn}}, \tag{13}$$

$$[k]_{de} = \frac{N_{k,de}}{V_{de}}, \tag{14}$$

$$[k]_{dg} = \frac{N_{k,dg}}{V_{dg}}. \tag{15}$$

We insert the Nernst-Planck equation for the axial flux density (Eq 1) into Eq 4 and get:

$$\frac{dN_{k,sn}}{dt} = -j_{k,msn}A_m + \frac{A_i D_k}{\lambda_i^2 \Delta x}\left[\gamma_k([k]_{dn} - [k]_{sn}) + \frac{z_k F}{RT}\overline{[k]}_n(\phi_{dn} - \phi_{sn})\right]. \tag{16}$$

In Eq 16, $[k]_{dn}$ and $[k]_{sn}$ are the intraneuronal ion concentrations of the dendrite and soma, defined in Eqs 13 and 10, respectively. We define the voltage variables $\phi_{dn}$ and $\phi_{sn}$ below.

**Six constraints to derive $\phi$.** If we have four ion species (Na$^+$, K$^+$, Cl$^-$, and Ca$^{2+}$) in six compartments, we get 24 equations to solve (Eqs 4–9 times four) and 30 unknowns ($N$ and $\phi$). We overcome this by defining $\phi$ in terms of ion concentrations using a set of constraints similar to those used in [33].

1. *Arbitrary reference point for $\phi$.* The first constraint is simple; we can choose an arbitrary reference point for $\phi$. We define it to be in the ECS of the dendrite layer, which gives us:

$$\phi_{de} = 0. \tag{17}$$

2. *Neuronal membrane is a capacitor (dendrite).* As the second constraint, we use that the membrane is a capacitor. This means that it will always separate a charge $Q$ on one side from an opposite charge $-Q$ on the other side. This gives rise to a voltage difference across the membrane

$$\phi_{mdn} = Q/C_m, \tag{18}$$

where $C_m$ is the total capacitance of the membrane, i.e., $C_m = c_m A_m$, where $c_m$ is the capacitance per membrane area. We know, by definition, that $\phi_{mdn} = \phi_{dn} - \phi_{de}$, and since $\phi_{de} = 0$, we get:

$$\phi_{mdn} = \phi_{dn} = \frac{Q_{dn}}{C_m}. \tag{19}$$

We assume bulk electroneutrality, meaning that all net charges in the dendritic

compartment must be on the membrane. It follows that $Q_{dn} = F\sum_k z_k[k]_{dn}V_{dn}$, where $F$ is

the Faraday constant, $z_k$ is the charge number of ion species $k$, $[k]_{dn}$ is the ion concentration, and $V_{dn}$ is the volume. By inserting this into Eq 19, we get

$$\phi_{dn} = (F\sum_k z_k[k]_{dn}V_{dn})/(c_m A_m).$$

(20)

3. *Neuronal membrane is a capacitor (soma).* The second constraint also applies to the soma, and gives us the criterion:

$$\phi_{sn} - \phi_{se} = \frac{Q_{sn}}{C_m} = (F\sum_k z_k[k]_{sn}V_{sn})/(c_m A_m).$$

(21)

Here, the outside potential is not set to zero, so this constraint is not sufficient to determine $\phi_{sn}$ and $\phi_{se}$ separately.

4. *Glial membrane is a capacitor (dendrite layer).* The glial membrane is no different than the neuronal membrane when it comes to acting as a capacitor, so we get:

$$\phi_{dg} = \frac{Q_{dg}}{C_m} = (F\sum_k z_k[k]_{dg}V_{dg})/(c_m A_m),$$

(22)

where we have used that $\phi_{de} = 0$.

5. *Glial membrane is a capacitor (soma layer).* Constraint number (4) also applies to the soma layer, and gives us:

$$\phi_{sg} - \phi_{se} = \frac{Q_{sg}}{C_m} = (F\sum_k z_k[k]_{sg}V_{sg})/(c_m A_m).$$

(23)

We can now calculate $\phi_{dn}$ and $\phi_{dg}$ from Eqs 20 and 22 but to determine $\phi_{sn}$, $\phi_{se}$, and $\phi_{sg}$, we need a sixth constraint.

6. *Current anti-symmetry.* The sixth constraint is charge (anti-)symmetry. We must define the initial conditions so that the membrane separates a charge $Q$ on one side from an opposite charge $-Q$ on the other side, and the system dynamics so that it stays this way. The membrane fluxes (alone) fulfill this criterion, since a charge that leaves a compartment automatically pops up on the other side of the membrane, making sure that $dQ_i/dt = -dQ_e/dt$. For the axial fluxes to fulfill the criterion, we must have that:

$$A_i i_{in} + A_i i_{ig} = -A_e i_e,$$

(24)

where $i$ stands for current density. We find expressions for $i_{in}$, $i_{ig}$, and $i_e$, by multiplying Eqs 1–3 by $Fz_k$ and sum over all ion species $k$. Expressions for the current densities then become:

$$i_{in} = -\frac{F}{\lambda_i^2 \Delta x}\sum_k D_k z_k \gamma_k([k]_{dn} - [k]_{sn}) - \frac{F^2}{RT\lambda_i^2 \Delta x}\sum_k D_k z_k^2 \overline{[k]}_n (\phi_{dn} - \phi_{sn}),$$

(25)

$$i_{ig} = -\frac{F}{\lambda_i^2 \Delta x}\sum_k D_k z_k([k]_{dg} - [k]_{sg}) - \frac{F^2}{RT\lambda_i^2 \Delta x}\sum_k D_k z_k^2 \overline{[k]}_g (\phi_{dg} - \phi_{sg}),$$

(26)

$$i_e = -\frac{F}{\lambda_e^2 \Delta x}\sum_k D_k z_k([k]_{de} - [k]_{se}) - \frac{F^2}{RT\lambda_e^2 \Delta x}\sum_k D_k z_k^2 \overline{[k]}_e (\phi_{de} - \phi_{se}).$$

(27)

The first term in Eq 25 is the diffusion current density and is defined by the ion

concentrations:

$$i_{\text{diff,in}} = -\frac{F}{\lambda_i^2 \Delta x} \sum_k D_k z_k \gamma_k ([k]_{\text{dn}} - [k]_{\text{sn}}). \tag{28}$$

The second term is the field driven current density

$$i_{\text{field,in}} = -\sigma_n \frac{(\phi_{\text{dn}} - \phi_{\text{sn}})}{\Delta x}, \tag{29}$$

where $\sigma_n$ is the conductivity:

$$\sigma_n = \frac{F^2}{RT\lambda_i^2} \sum_k D_k z_k^2 \overline{[k]}_n. \tag{30}$$

Likewise, Eq 26 can be written in terms of $i_{\text{diff,ig}}$, $i_{\text{field,ig}}$, and $\sigma_g$, and Eq 27 in terms of $i_{\text{diff,e}}$, $i_{\text{field,e}}$, and $\sigma_e$. We combine Eqs 24–27 and obtain:

$$-A_i i_{\text{diff,in}} + A_i \sigma_n \frac{(\phi_{\text{dn}} - \phi_{\text{sn}})}{\Delta x} - A_i i_{\text{diff,ig}} + A_i \sigma_g \frac{(\phi_{\text{dg}} - \phi_{\text{sg}})}{\Delta x} =$$
$$A_e i_{\text{diff,e}} - A_e \sigma_e \frac{(\phi_{\text{de}} - \phi_{\text{se}})}{\Delta x}. \tag{31}$$

In Eq 31, we know $\phi_{\text{dn}}$, $\phi_{\text{de}}$, and $\phi_{\text{dg}}$ from Eqs 20, 17, and 22, and $i_{\text{diff}}$ and $\sigma$ from the ion concentrations. We solve Eqs 21, 23, and 31 to find $\phi_{\text{sn}}$, $\phi_{\text{sg}}$, and $\phi_{\text{se}}$:

$$\phi_{\text{se}} = \quad (-\Delta x A_i i_{\text{diff,in}} + A_i \sigma_n \phi_{\text{dn}} - A_i \sigma_n \frac{Q_{\text{sn}}}{c_m A_m} - \Delta x A_i i_{\text{diff,ig}} \tag{32}$$

$$+ A_i \sigma_g \phi_{\text{dg}} - A_i \sigma_g \frac{Q_{\text{sg}}}{c_m A_m} - \Delta x A_e i_{\text{diff,e}}) \tag{33}$$

$$/ (A_e \sigma_e + A_i \sigma_n + A_i \sigma_g), \tag{34}$$

$$\phi_{\text{sn}} = \quad \frac{Q_{\text{sn}}}{c_m A_m} + \phi_{\text{se}}, \tag{35}$$

$$\phi_{\text{sg}} = \quad \frac{Q_{\text{sg}}}{c_m A_m} + \phi_{\text{se}}. \tag{36}$$

## Neuronal membrane mechanisms

The neuronal membrane mechanisms were the same as in [33], where the active ion channels were taken from the Pinsky-Rinzel model [43], leak currents, ion pumps, and cotransporters were modeled as in [59], and a 2Na$^+$/Ca$^{2+}$ exchanger was added to partly mimic the Ca$^{2+}$ decay in [43]. We list the mechanisms again here for easy reference.

## Leakage channels

Both neuronal compartments contained $Na^+$, $K^+$, and $Cl^-$ leak currents. The flux densities were modeled as follows:

$$j_{k,leak} = \overline{g}_{k,leak}(\phi_m - E_k)/(Fz_k),$$
(37)

where $k$ denotes the ion species, $\overline{g}_{k,leak}$ is the ion conductance, $\phi_m$ is the membrane potential, $E_k$ is the reversal potential, $F$ is the Faraday constant, and $z_k$ is the charge number. Reversal potentials are given by the Nernst equation:

$$E_k = \frac{RT}{z_k F} \ln \frac{[k]_e}{\gamma_k [k]_i},$$
(38)

where $R$ is the gas constant, $T$ is the absolute temperature, $\gamma_k$ is the intracellular fraction of freely moving ions, and $[k]_e$ and $[k]_i$ are the extra- and intracellular concentrations of ion species $k$, respectively.

## Active ion channels

The active ion channels included $Na^+$ and $K^+$ delayed rectifier fluxes in the soma ($j_{Na}$, $j_{DR}$), and a voltage-dependent $Ca^{2+}$ flux ($j_{Ca}$), a voltage-dependent $K^+$ afterhyperpolarization flux ($j_{AHP}$), and a $Ca^{2+}$-dependent $K^+$ flux ($j_C$) in the dendrite:

$$j_{Na} = g_{Na}(\phi_{msn} - E_{Na,sn})/(Fz_{Na}),$$
(39)

$$j_{DR} = g_{DR}(\phi_{msn} - E_{K,sn})/(Fz_K),$$
(40)

$$j_{Ca} = g_{Ca}(\phi_{mdn} - E_{Ca,dn})/(Fz_{Ca}),$$
(41)

$$j_{AHP} = g_{AHP}(\phi_{mdn} - E_{K,dn})/(Fz_K),$$
(42)

$$j_C = g_C(\phi_{mdn} - E_{K,dn})/(Fz_K).$$
(43)

Here, $g_{Na}$, $g_{DR}$, $g_{Ca}$, $g_{AHP}$, and $g_C$ are ion conductances, $\phi_{msn}$ and $\phi_{mdn}$ are the somatic and dendritic membrane potentials, respectively, $E_{Na,sn}$, $E_{K,sn}$, $E_{Ca,dn}$, and $E_{K,dn}$ are reversal potentials, $F$ is the Faraday constant, and $z_{Na}$, $z_K$, and $z_{Ca}$ are charge numbers. We used the Hodkin-Huxley formalism to model the voltage-dependent conductances, with differential equations for the gating variables:

$$\frac{dx}{dt} = \alpha_x(1 - x) - \beta_x x, \quad \text{with } x \in \{m, h, n, s, c, q\},$$
(44)

$$\frac{dz}{dt} = \frac{z_\infty - z}{\tau_z},$$
(45)

and

$$g_{\text{Na}} = \overline{g}_{\text{Na}} m_\infty^2 h, \tag{46}$$

$$g_{\text{DR}} = \overline{g}_{\text{DR}} n, \tag{47}$$

$$g_{\text{Ca}} = \overline{g}_{\text{Ca}} s^2 z, \tag{48}$$

$$g_{\text{C}} = \overline{g}_{\text{C}} c \chi([\text{Ca}^{+2}]_{\text{dn}}), \tag{49}$$

$$g_{\text{AHP}} = \overline{g}_{\text{AHP}} q, \tag{50}$$

$$\alpha_{\text{m}} = -\frac{3.2 \cdot 10^5 \cdot \phi_1}{\exp\left(-\phi_1/0.004\right) - 1}, \text{ with } \phi_1 = \phi_{\text{msn}} + 0.0469 \tag{51}$$

$$\beta_{\text{m}} = \frac{2.8 \cdot 10^5 \cdot \phi_2}{\exp\left(\phi_2/0.005\right) - 1}, \text{ with } \phi_2 = \phi_{\text{msn}} + 0.0199 \tag{52}$$

$$m_\infty = \frac{\alpha_{\text{m}}}{\alpha_{\text{m}} + \beta_{\text{m}}} \tag{53}$$

$$\alpha_{\text{h}} = 128 \exp\frac{-0.043 - \phi_{\text{msn}}}{0.018}, \tag{54}$$

$$\beta_{\text{h}} = \frac{4000}{1 + \exp\left(-\phi_3/0.005\right)}, \text{ with } \phi_3 = \phi_{\text{msn}} + 0.02 \tag{55}$$

$$\alpha_{\text{n}} = -\frac{1.6 \cdot 10^4 \cdot \phi_4}{\exp\left(-\phi_4/0.005\right) - 1}, \text{ with } \phi_4 = \phi_{\text{msn}} + 0.0249 \tag{56}$$

$$\beta_{\text{n}} = 250 \exp\left(-\phi_5/0.04\right), \text{ with } \phi_5 = \phi_{\text{msn}} + 0.04 \tag{57}$$

$$\alpha_{\text{s}} = \frac{1600}{1 + \exp\left(-72(\phi_{\text{mdn}} - 0.005)\right)}, \tag{58}$$

$$\beta_{\text{s}} = \frac{2 \cdot 10^4 \cdot \phi_6}{\exp\left(\phi_6/0.005\right) - 1}, \text{ with } \phi_6 = \phi_{\text{mdn}} + 0.0089 \tag{59}$$

$$z_\infty = \frac{1}{1 + \exp\left(\phi_7/0.001\right)}, \text{ with } \phi_7 = \phi_{\text{mdn}} + 0.03 \tag{60}$$

$$\tau_z = 1, \tag{61}$$

$$\alpha_{\text{c}} = \begin{cases} 52.7 \exp\left(\frac{\phi_8}{0.011} - \frac{\phi_9}{0.027}\right), & \text{if } \phi_{\text{mdn}} \leq -0.01 \text{ V} \\ 2000 \exp\left(-\phi_9/0.027\right), & \text{otherwise} \end{cases} \tag{62}$$

$$\text{with } \phi_8 = \phi_{\text{mdn}} + 0.05 \text{ and } \phi_9 = \phi_{\text{mdn}} + 0.0535 \tag{63}$$

$$\beta_{\text{c}} = \begin{cases} 2000 \exp\left(-\phi_9/0.027\right) - \alpha_{\text{c}}, & \text{if } \phi_{\text{mdn}} \leq -0.01 \text{ V} \\ 0, & \text{otherwise} \end{cases} \tag{64}$$

$$\chi = \min\left(\frac{\gamma_{\text{Ca}}[\text{Ca}^{+2}]_{\text{dn}} - 99.8 \cdot 10^{-6}}{2.5 \cdot 10^{-4}}, 1\right), \tag{65}$$

$$\alpha_{\text{q}} = \min(2 \cdot 10^4 (\gamma_{\text{Ca}}[\text{Ca}^{+2}]_{\text{dn}} - 99.8 \cdot 10^{-6}), 10), \tag{66}$$

$$\beta_{\text{q}} = 1. \tag{67}$$

In Eqs 44–67, rates ($\alpha$'s, $\beta$'s) are in units of 1/s, $\tau_z$ is in units of s, and voltages $\phi$ are in units of V.

## Stabilizing mechanisms

Both neuronal compartments contained a 3Na$^+$/2K$^+$ pump, a K$^+$/Cl$^-$ cotransporter (KCC2), a Na$^+$/K$^+$/2Cl$^-$ cotransporter (NKCC1), and a 2Na$^+$/Ca$^{2+}$ exchanger:

$$j_{\text{pump,n}} = \frac{\rho_{\text{n}}}{1.0 + \exp\left((25 - [\text{Na}^+]_{\text{n}})/3\right)} \cdot \frac{1.0}{1.0 + \exp\left(3.5 - [\text{K}^+]_{\text{e}}\right)}, \tag{68}$$

$$j_{\text{kcc2}} = U_{\text{kcc2}} \ln\left(\frac{[\text{K}^+]_{\text{n}}[\text{Cl}^-]_{\text{n}}}{[\text{K}^+]_{\text{e}}[\text{Cl}^-]_{\text{e}}}\right), \tag{69}$$

$$j_{\text{nkcc1}} = U_{\text{nkcc1}} f([\text{K}^+]_{\text{e}}) \left( \ln\left(\frac{[\text{K}^+]_{\text{n}}[\text{Cl}^-]_{\text{n}}}{[\text{K}^+]_{\text{e}}[\text{Cl}^-]_{\text{e}}}\right) + \ln\left(\frac{[\text{Na}^+]_{\text{n}}[\text{Cl}^-]_{\text{n}}}{[\text{Na}^+]_{\text{e}}[\text{Cl}^-]_{\text{e}}}\right) \right), \tag{70}$$

$$f([\text{K}^+]_{\text{e}}) = \frac{1}{1 + \exp\left(16 - [\text{K}^+]_{\text{e}}\right)}, \tag{71}$$

$$j_{\text{Ca-dec}} = U_{\text{Ca-dec}} \cdot ([\text{Ca}^{+2}]_{\text{n}} - [\text{Ca}^{+2}]_{\text{n,b}}) \cdot \frac{V_{\text{n}}}{A_{\text{m}}}. \tag{72}$$

Here, $n$ denotes the neuronal compartment, $e$ denotes the extracellular compartment, $\rho_{\text{n}}$, $U_{\text{kcc2}}$, and $U_{\text{nkcc1}}$ are pump and cotransporter strengths, $U_{\text{Ca-dec}}$ is the Ca$^{2+}$ decay rate, and $[\text{Ca}^{+2}]_{\text{n,b}}$ is the basal Ca$^{2+}$ concentration.

## Glial membrane mechanisms

The glial membrane mechanisms were taken from a previously published astrocyte model [47]. They included Na$^+$ and Cl$^-$ leak channels, modeled as in Eq 37, an inward rectifying K$^+$ channel, and a 3Na$^+$/2K$^+$ pump:

$$j_{\text{K-IR}} = \overline{g}_{\text{K-IR}} f_{\text{K-IR}} (\phi_{\text{mg}} - E_{\text{K,g}})/(Fz_{\text{K}}), \tag{73}$$

$$f_{K-IR} = \sqrt{\frac{[K^+]_e}{[K^+]_{e,b}}} \left( \frac{1 + \exp(18.4/42.4)}{1 + \exp((\Delta\phi \cdot 1000 + 18.5)/42.5)} \right)$$
$$\cdot \left( \frac{1 + \exp(-(118.6 + E_{K,b} \cdot 1000)/44.1)}{1 + \exp(-(118.6 + \phi_{mg} \cdot 1000)/44.1)} \right) \tag{74}$$

$$j_{pump,g} = \rho_g \frac{[Na^+]_g^{1.5}}{[Na^+]_g^{1.5} + [Na^+]_{g,threshold}^{1.5}} \frac{[K^+]_e}{[K^+]_e + [K^+]_{e,threshold}}. \tag{75}$$

Here, $g$ denotes the glial compartment, $e$ denotes the extracellular compartment, $\overline{g}_{K-IR}$ is the $K^+$ ion conductance, $\phi_{mg}$ is the membrane potential, $E_{K,g}$ is the $K^+$ reversal potential, $F$ is the Faraday constant, $z_K$ is the $K^+$ charge number, $[K^+]_{e,b}$ is the basal $K^+$ concentration in the extracellular space, $\Delta\phi = \phi_{mg} - E_K$, $E_{K,b}$ is the reversal potential for $K^+$ at basal concentrations, $\rho_g$ is the pump strength, and $[Na^+]_{g,threshold}$ and $[K^+]_{e,threshold}$ are the pump's threshold concentrations for $Na^+$ and $K^+$, respectively. We included the same set of membrane mechanisms in both glial compartments.

## Volume dynamics

To calculate the osmotically induced volume changes $dV/dt$, we used the formalism outlined in [51]. The water flow $Q$ across the membrane is given by

$$Q = G\Delta\Psi, \tag{76}$$

where $G$ is the water permeability, given in units of m³/Pa/s, and $\Delta\Psi$ is the pressure difference between the inside ($i$) and the outside ($e$) of the cell, $\Psi_i - \Psi_e$, given in units of Pa. We assumed the hydrostatic pressure differences to be zero, so that water flow was driven by osmotic pressure differences only, and we calculated the solute potentials from:

$$\Psi = -iMRT. \tag{77}$$

Here, $i$ is the ionization factor (van't Hoff factor), which is 1 for ions, $M$ is the osmotic concentration of solutes measured in moles per cubic meter, $R$ is the gas constant, and $T$ is the absolute temperature. If we combine Eqs 76 and 77 and notice that $Q = -dV_i/dt$, where $V_i$ is the intracellular volume, the osmotically induced volume changes are given by the following differential equations:

$$\frac{dV_{sn}}{dt} = -G_n RT \left( \frac{N_{tot,se}}{V_{se}} - \frac{N_{tot,sn}}{V_{sn}} \right), \tag{78}$$

$$\frac{dV_{sg}}{dt} = -G_g RT \left( \frac{N_{tot,se}}{V_{se}} - \frac{N_{tot,sg}}{V_{sg}} \right), \tag{79}$$

$$\frac{dV_{dn}}{dt} = -G_n RT \left( \frac{N_{tot,de}}{V_{de}} - \frac{N_{tot,dn}}{V_{dn}} \right), \tag{80}$$

$$\frac{dV_{dg}}{dt} = -G_g RT \left( \frac{N_{tot,de}}{V_{de}} - \frac{N_{tot,dg}}{V_{dg}} \right), \tag{81}$$

$$\frac{dV_{se}}{dt} = -\left(\frac{dV_{sn}}{dt} + \frac{dV_{sg}}{dt}\right), \tag{82}$$

$$\frac{dV_{de}}{dt} = -\left(\frac{dV_{dn}}{dt} + \frac{dV_{dg}}{dt}\right), \tag{83}$$

where $N_{tot}$ is the total amount of ions in a compartment, given in moles. Eqs 82 and 83 follow from the assumption that the total volume did not change, that is, the system was closed.

We only considered effects of transmembrane water flow, and intra-domain water flow due to hydrostatic pressures were neglected. The assumption is applied in several previous multicompartment models, e.g. [37, 39, 83], but as Mori et al. [37] point out, intra-domain water flow in brain tissue is not fully understood and may play an important role in brain function. It is an ongoing area of computational research, see, e.g. [113–115], and the edNEG model may contribute to this work if the framework gets extended.

## Model summary

To keep track of all ions in the system, we solved six differential equations for each ion species $k$:

$$\frac{dN_{k,sn}}{dt} = -j_{k,msn}A_m - j_{k,in}A_i, \tag{84}$$

$$\frac{dN_{k,se}}{dt} = +j_{k,msn}A_m - j_{k,e}A_e + j_{k,msg}A_m, \tag{85}$$

$$\frac{dN_{k,sg}}{dt} = -j_{k,msg}A_m - j_{k,ig}A_i, \tag{86}$$

$$\frac{dN_{k,dn}}{dt} = -j_{k,mdn}A_m + j_{k,in}A_i, \tag{87}$$

$$\frac{dN_{k,de}}{dt} = +j_{k,mdn}A_m + j_{k,e}A_e + j_{k,mdg}A_m, \tag{88}$$

$$\frac{dN_{k,dg}}{dt} = -j_{k,mdg}A_m + j_{k,ig}A_i. \tag{89}$$

The total membrane flux densities are summarized here:

$$j_{Na,msn} = j_{Na} + j_{Na,leak,n} + 3j_{pump,n} + j_{nkcc1} - 2j_{Ca-dec}, \tag{90}$$

$$j_{K,msn} = j_{DR} + j_{K,leak,n} - 2j_{pump,n} + j_{nkcc1} + j_{kcc2}, \tag{91}$$

$$j_{Cl,msn} = j_{Cl,leak,n} + 2j_{nkcc1} + j_{kcc2}, \tag{92}$$

$$j_{Ca,msn} = j_{Ca-dec}, \tag{93}$$

$$j_{Na,mdn} = j_{Na,leak,n} + 3j_{pump,n} + j_{nkcc1} - 2j_{Ca-dec}, \tag{94}$$

$$j_{K,mdn} = j_{AHP} + j_C + j_{K,leak,n} - 2j_{pump,n} + j_{nkcc1} + j_{kcc2}, \tag{95}$$

$$j_{Cl,mdn} = j_{Cl,leak,n} + 2j_{nkcc1} + j_{kcc2}, \tag{96}$$

$$j_{Ca,mdn} = j_{Ca} + j_{Ca-dec}, \tag{97}$$

$$j_{Na,msg} = j_{Na,leak,g} + 3j_{pump,g}, \tag{98}$$

$$j_{K,msg} = j_{K-IR} - 2j_{pump,g}, \tag{99}$$

$$j_{Cl,msg} = j_{Cl,leak,g}, \tag{100}$$

$$j_{Na,mdg} = j_{Na,leak,g} + 3j_{pump,g}, \tag{101}$$

$$j_{K,mdg} = j_{K-IR} - 2j_{pump,g}, \tag{102}$$

$$j_{Cl,mdg} = j_{Cl,leak,g}. \tag{103}$$

At each time step, we derived $\phi$ algebraically in all six compartments:

$$\phi_{de} = 0, \tag{104}$$

$$\phi_{dn} = (F\sum_k z_k [k]_{dn} V_{dn})/(c_m A_m), \tag{105}$$

$$\phi_{dg} = (F\sum_k z_k [k]_{dg} V_{dg})/(c_m A_m), \tag{106}$$

$$\phi_{se} = (-\Delta x A_i i_{diff,in} + A_i \sigma_n \phi_{dn} - A_i \sigma_n \frac{Q_{sn}}{c_m A_{sn}} - \Delta x A_i i_{diff,ig} \tag{107}$$

$$+ A_i \sigma_g \phi_{dg} - A_i \sigma_g \frac{Q_{sg}}{c_m A_m} - \Delta x A_e i_{diff,e}) \tag{108}$$

$$/(A_e \sigma_e + A_i \sigma_n + A_i \sigma_g), \tag{109}$$

$$\phi_{sn} = \frac{Q_{sn}}{c_m A_m} + \phi_{se}, \tag{110}$$

$$\phi_{sg} = \frac{Q_{sg}}{c_m A_m} + \phi_{se}. \tag{111}$$

Membrane potentials were defined as:

$$\phi_{\text{msn}} = \phi_{\text{sn}} - \phi_{\text{se}}, \tag{112}$$

$$\phi_{\text{mdn}} = \phi_{\text{dn}}, \tag{113}$$

$$\phi_{\text{msg}} = \phi_{\text{sg}} - \phi_{\text{se}}, \tag{114}$$

$$\phi_{\text{mdg}} = \phi_{\text{dg}}. \tag{115}$$

Volume dynamics was given by:

$$\frac{dV_{\text{sn}}}{dt} = -G_{\text{n}}RT\left(\frac{N_{\text{tot,se}}}{V_{\text{se}}} - \frac{N_{\text{tot,sn}}}{V_{\text{sn}}}\right), \tag{116}$$

$$\frac{dV_{\text{sg}}}{dt} = -G_{\text{g}}RT\left(\frac{N_{\text{tot,se}}}{V_{\text{se}}} - \frac{N_{\text{tot,sg}}}{V_{\text{sg}}}\right), \tag{117}$$

$$\frac{dV_{\text{dn}}}{dt} = -G_{\text{n}}RT\left(\frac{N_{\text{tot,de}}}{V_{\text{de}}} - \frac{N_{\text{tot,dn}}}{V_{\text{dn}}}\right), \tag{118}$$

$$\frac{dV_{\text{dg}}}{dt} = -G_{\text{g}}RT\left(\frac{N_{\text{tot,de}}}{V_{\text{de}}} - \frac{N_{\text{tot,dg}}}{V_{\text{dg}}}\right), \tag{119}$$

$$\frac{dV_{\text{se}}}{dt} = -\left(\frac{dV_{\text{sn}}}{dt} + \frac{dV_{\text{sg}}}{dt}\right), \tag{120}$$

$$\frac{dV_{\text{de}}}{dt} = -\left(\frac{dV_{\text{dn}}}{dt} + \frac{dV_{\text{dg}}}{dt}\right). \tag{121}$$

Fig 2 summarizes the model and model parameters are listed in Tables 1–5.

## Simulations

### Model tuning

The edNEG model combines two previous models, one consisting of a neuron and ECS [33], and the other of a glial domain (astrocyte) and ECS [47]. When we combined the models, we set the initial concentrations in the glial domain to the same values as in [47]. In the ECS, we set the initial Na$^+$, K$^+$, and Cl$^-$ concentrations to the same values as in [47], and the initial Ca$^{2+}$

**Table 3. Temperature and physical constants[†].**

| Parameter | Value | Reference |
|---|---|---|
| $T$ (absolute temperature) | 309.14K | [33] |
| $F$ (Faraday constant) | $9.648 \cdot 10^4$ C/mol | |
| $R$ (gas constant) | 8.314 J/(mol K) | |

[†] This table is adopted from [33].

**Table 4. Membrane parameters.**

| Parameter | Value | Reference |
|---|---|---|
| $c_m$ | $3 \cdot 10^{-2}$ F/m$^2$ | [33] |
| $\bar{g}_{Na,leak,n}$ | 0.246 S/m$^2$ | Eq 123 |
| $\bar{g}_{K,leak,n}$ | 0.245 S/m$^2$ | Eq 123 |
| $\bar{g}_{Cl,leak,n}$ | 1 S/m$^2$ | [33] |
| $\bar{g}_{Na}$ | 300 S/m$^2$ | [33] |
| $\bar{g}_{DR}$ | 150 S/m$^2$ | [33] |
| $\bar{g}_{Ca}$ | 118 S/m$^2$ | [33] |
| $\bar{g}_{AHP}$ | 8 S/m$^2$ | [33] |
| $\bar{g}_{C}$ | 150 S/m$^2$ | [33] |
| $\rho_n$ | $1.87 \cdot 10^{-6}$ mol/(m$^2$s) | [33] |
| $U_{kcc2}$ | $1.49 \cdot 10^{-7}$ mol/(m$^2$s) | Eq 124 |
| $U_{nkcc1}$ | $2.33 \cdot 10^{-7}$ mol/(m$^2$s) | [33] |
| $U_{Ca-dec}$ | 75 s$^{-1}$ | [33] |
| $\bar{g}_{Na,leak,g}$ | 1 S/m$^2$ | [47] |
| $\bar{g}_{Cl,leak,g}$ | 0.5 S/m$^2$ | [47] |
| $\bar{g}_{K-IR}$ | 16.96 S/m$^2$ | [47] |
| $\rho_g$ | $1.12 \cdot 10^{-6}$ mol/(m$^2$s) | [47] |
| $[Na^+]_{g,threshold}$ | 10 mM | [47] |
| $[K^+]_{e,threshold}$ | 1.5 mM | [47] |
| $G_n$ | $2 \cdot 10^{-23}$ m$^3$/Pa/s | [116] |
| $G_g$ | $5 \cdot 10^{-23}$ m$^3$/Pa/s | [50] |

**Table 5. Initial conditions.**

| Variables | Pre-calibrated | Post-calibrated[1] | Reference |
|---|---|---|---|
| $\phi_{mn,0}$ [†] | −67.7 mV | −66.9 mV | [33] |
| $\phi_{mg,0}$ [†] | −83.6 mV | −83.9 mV | [47] |
| $[Na^+]_{n,0}$ | 16.9 mM | 18.7 mM | [33] |
| $[Na^+]_{e,0}$ | 144.622 mM | 142.3 mM | [47] |
| $[Na^+]_{g,0}$ | 15.189 mM | 14.5 mM | [47] |
| $[K^+]_{n,0}$ | 139.5 mM | 138.1 mM | [33] |
| $[K^+]_{e,0}$ | 3.082 mM | 3.5 mM | [47] |
| $[K^+]_{g,0}$ | 99.959 mM | 101.2 mM | [47] |
| $[Cl^-]_{n,0}$ | 6.7412 mM | 7.1 mM | Eq 122 |
| $[Cl^-]_{e,0}$ | 133.71 mM | 131.9 mM | [47] |
| $[Cl^-]_{g,0}$ | 5.145 mM | 5.7 mM | [47] |
| $[Ca^{2+}]_{n,0}$ | 0.01 mM[*] | 0.01 mM[*] | [33] |
| $[Ca^{2+}]_{e,0}$ | 1.1 mM | 1.1 mM | [33] |
| $n_0$ | 0.0003 | 0.0003 | [33] |
| $h_0$ | 0.999 | 0.9993 | [33] |
| $s_0$ | 0.007 | 0.0077 | [33] |
| $c_0$ | 0.005 | 0.0057 | [33] |
| $q_0$ | 0.011 | 0.0117 | [33] |
| $z_0$ | 1.0 | 1.0 | [33] |

[1] Values with more decimals included were read to/from file and used in the simulations. (Available at https://github.com/CINPLA/edNEGmodel_analysis.)

[†] $\phi_m$ is not an independent state variable, but defined at each time point from the ion concentrations.

[*] Only 1% of the total intracellular Ca$^{2+}$, that is, a 100 nM, was assumed to be free (unbuffered).

concentration to the same value as in [33]. In the neuron, we set the initial $Na^+$, $K^+$, and $Ca^{2+}$ concentrations to be the same as in [33]. If we used the same $Cl^-$ concentration as in [33], we obtained an unrealistically low reversal potential for $Cl^-$. Therefore, we computed a new value for the intraneuronal $Cl^-$ concentration by requiring the initial reversal potential for $Cl^-$ to be the same as in [33], i.e., we solved:

$$\frac{RT}{z_{Cl}F} \ln \frac{[Cl^-]_{e,new}}{\gamma_{Cl}[Cl^-]_{n,new}} = \frac{RT}{z_{Cl}F} \ln \frac{[Cl^-]_{e,old}}{\gamma_{Cl}[Cl^-]_{n,old}}, \qquad (122)$$

where $[Cl^-]_{n,old}$ $[Cl^-]_{e,old}$ are the intra- and extraneuronal $Cl^-$ concentrations at steady state in [33], $[Cl^-]_{e,new}$ is the ECS $Cl^-$ concentration adopted from [47] (Table 5), and $[Cl^-]_{n,new}$ is the new intraneuronal $Cl^-$ concentration computed for the edNEG model (Table 5).

As the initial ion concentrations (Table 5) differed from the initial ECS- and intraneuronal $Cl^-$ concentrations in the previous neuron model [33], the neuron was not in equilibrium with the (new) environment. This was because the altered ion concentrations gave rise to altered concentration-dependent activity of the ion pumps, cotransporters, and ionic currents through $Na^+$ and $K^+$ channels. We found that a re-tuning of the $Na^+$ and $K^+$ leak conductances ($\bar{g}_{Na,leak,n}$ and $\bar{g}_{K,leak,n}$) and $K^+/Cl^-$ cotransporter strength ($U_{kcc2}$) in the neuron model was sufficient to obtain a system with a plausible resting state. Tuning the leak conductances was done by requiring that the initial leakage currents should be identical to those in [33], i.e., we set:

$$\bar{g}_{k,new}(\phi_m - E_{k,new}) = \bar{g}_{k,old}(\phi_m - E_{k,old}), \qquad (123)$$

with $\phi_m$ being the resting potential in [33] (−67.7 mV), $E_{k,old}$ being the reversal potential for ion species $k$ at steady state in [33], and $E_{k,new}$ being the reversal potential obtained by the new initial ion concentrations (Table 5). Similarly, we tuned the $K^+/Cl^-$ cotransporter strength by requiring the initial $K^+$ and $Cl^-$ currents through the cotransporter to be identical to those in [33], i.e., we set:

$$U_{kcc2,new} \ln \left( \frac{[K^+]_{n,new}[Cl^-]_{n,new}}{[K^+]_{e,new}[Cl^-]_{e,new}} \right) = U_{kcc2,old} \ln \left( \frac{[K^+]_{n,old}[Cl^-]_{n,old}}{[K^+]_{e,old}[Cl^-]_{e,old}} \right), \qquad (124)$$

with $[k]_{n,new}$ and $[k]_{e,new}$ being the new neuronal and extracellular ion concentrations of ion species $k$, respectively, and $[k]_{n,old}$ and $[k]_{e,old}$ being the neuronal and extracellular ion concentrations at steady state in [33]. By solving Eqs 123 and 124, we obtained a final set of passive conductances and cotransporter strengths for the neuronal membrane (Table 4).

In the previous neuron model [33], the extracellular redistribution of ions was very fast, and almost no concentration gradients developed between the soma and dendrite layer, even during intense neural activity. To obtain concentration gradients and diffusion potentials more consistent with that seen in experiments provoking intense neural activity (see, e.g., [8, 13]), we reduced the cross-section area for extracellular fluxes (and currents) by a factor 10 compared to the previous model (Table 1).

After calibrating the edNEG model (running it for 5000 s) with the (new) derived passive conductances and $K^+/Cl^-$ cotransporter strength, it settled at a resting state where the neuronal resting membrane potential was −66.9 mV, and the glial membrane resting potential was −83.9 mV, which were close to the original resting potentials for the neuron and glial domain (original values were −67.7 mV [33] and −83.6 mV [47], respectively).

## Initial conditions

Before tuning the edNEG model, we defined its initial volumes (Table 1), amounts of ions, membrane potentials, and gating variables (Table 5, Pre-calibrated column) using values from the two previous models in [33] and [47] and Eq 122. After re-tuning selected parameters (as described in the previous subsection), the system was close to, but not strictly in equilibrium, and for this reason we calibrated the edNEG model by running it for 5000 s of simulated time. The water permeabilities were set to zero during the calibration.

We wrote the final values from the calibration to file (see Table 5, Post-calibrated column) and used them as initial conditions in all simulations shown throughout this paper. Note that the edNEG model takes amounts of ions (in units of mol) as input, while we have listed ion concentrations in Table 5. The post-calibrated values of the ion concentrations correspond to the following reversal potentials: $E_{\text{Na,n}} = 54$ mV, $E_{\text{Na,g}} = 61$ mV, $E_{\text{K,n}} = -98$ mV, $E_{\text{K,g}} = -89$ mV, $E_{\text{Cl,n}} = -78$ mV, $E_{\text{Cl,g}} = -84$ mV, and $E_{\text{Ca,n}} = 124$ mV.

To ensure charge symmetry and electroneutrality, we defined a set of static residual charges, based on the initial amounts of ions. These represent negatively charged macromolecules present in real cells. We defined them as constant amounts of ion species $X^-$ with charge number $z_X = -1$ and diffusion constant $D_X = 0$. To ensure strict electroneutrality, we did not read residual charges to/from file, but calculated them at the beginning of each simulation. They were given by the following expressions:

$$N_{\text{X,n}} = z_{\text{Na}}N_{\text{Na,n,0}} + z_{\text{K}}N_{\text{K,n,0}} + z_{\text{Cl}}N_{\text{Cl,n,0}} + z_{\text{Ca}}N_{\text{Ca,n,0}} - \phi_{\text{mn,0}}\frac{c_{\text{m}}A_{\text{m}}}{F}, \tag{125}$$

$$N_{\text{X,e}} = z_{\text{Na}}N_{\text{Na,e,0}} + z_{\text{K}}K_{\text{K,e,0}} + z_{\text{Cl}}N_{\text{Cl,e,0}} + z_{\text{Ca}}N_{\text{Ca,e,0}} + (\phi_{\text{mn,0}} + \phi_{\text{mg,0}})\frac{c_{\text{m}}A_{\text{m}}}{F}, \tag{126}$$

$$N_{\text{X,g}} = z_{\text{Na}}N_{\text{Na,g,0}} + z_{\text{K}}N_{\text{K,g,0}} + z_{\text{Cl}}N_{\text{Cl,g,0}} - \phi_{\text{mg,0}}\frac{c_{\text{m}}A_{\text{m}}}{F}. \tag{127}$$

Additionally, we introduced a set of static residual molecules to ensure zero osmotic pressure gradients across the membranes at the beginning of each simulation. These were defined as osmotic concentrations of a molecule M:

$$[M]_{\text{n}} = (N_{\text{Na,n,0}} + N_{\text{K,n,0}} + N_{\text{Cl,n,0}} + N_{\text{Ca,n,0}})/V_{\text{sn,0}}, \tag{128}$$

$$[M]_{\text{e}} = (N_{\text{Na,e,0}} + N_{\text{K,e,0}} + N_{\text{Cl,e,0}} + N_{\text{Ca,e,0}})/V_{\text{se,0}}, \tag{129}$$

$$[M]_{\text{g}} = (N_{\text{Na,g,0}} + N_{\text{K,g,0}} + N_{\text{Cl,g,0}})/V_{\text{sg,0}}. \tag{130}$$

## Injection current

In Figs 3–6, we stimulated the neuron by applying a $K^+$, $Na^+$, or $Cl^-$ injection current into the somatic, dendritic, or both compartments. To ensure ion conservation, the same amount of ions were removed from the extracellular compartment(s). In this aspect, the stimulus was equivalent to a current through an open ion channel. This changed the ion dynamics in the

following way:

$$\frac{dN_{k,i}}{dt} \quad \rightarrow \quad \frac{dN_{k,i}}{dt} + \frac{I_{stim}}{Fz_k}, \tag{131}$$

$$\frac{dN_{k,e}}{dt} \quad \rightarrow \quad \frac{dN_{k,e}}{dt} - \frac{I_{stim}}{Fz_k}, \tag{132}$$

where $N_k$ is the amount of ions, $k$ denotes the injected ion species, $i$ denotes the relevant intra-cellular compartment, $e$ denotes the corresponding extracellular compartment, $F$ is the Faraday constant, $z_k$ is the charge number of ion species $k$, and $I_{stim}$ is the injection current, given in units of A. If nothing else is stated, we applied a $K^+$ injection current to the soma. A previous computational study of a cardiac cell showed that $K^+$ ions cause least physiological disruption when used as stimulus [68].

## Synaptic current

In Fig 7, we stimulated the neuron by attaching an AMPA synapse to the somatic, dendritic, or both compartments. We made the synaptic current ion-specific and modeled the synaptic conductance changes using a dual exponential function:

$$I_{syn,Na} \quad = \overline{g}_{syn,Na} \sum_{s=1}^{n} \left( \exp\left(-\frac{t-t_s}{\tau_1}\right) - \exp\left(-\frac{t-t_s}{\tau_2}\right) \right) \Theta(t-t_s)(\phi_m - E_{Na}), \tag{133}$$

$$I_{syn,K} \quad = \overline{g}_{syn,K} \sum_{s=1}^{n} \left( \exp\left(-\frac{t-t_s}{\tau_1}\right) - \exp\left(-\frac{t-t_s}{\tau_2}\right) \right) \Theta(t-t_s)(\phi_m - E_{K}), \tag{134}$$

$$I_{syn,Ca} \quad = \overline{g}_{syn,Ca} \sum_{s=1}^{n} \left( \exp\left(-\frac{t-t_s}{\tau_1}\right) - \exp\left(-\frac{t-t_s}{\tau_2}\right) \right) \Theta(t-t_s)(\phi_m - E_{Ca}). \tag{135}$$

In Eqs 133–135, $\Theta(t)$ is the Heaviside unit-step function: $\Theta(t \geq 0) = 1$, $\Theta(t < 0) = 0$, $\overline{g}_{syn,Na}$, $\overline{g}_{syn,K}$, and $\overline{g}_{syn,Ca}$ are the maximum $Na^+$, $K^+$, and $Ca^{2+}$ conductances, respectively, $t$ is time, $t_s$ is the arrival time of the $s$th spike, $\tau_1$ is the decay time constant, $\tau_2$ is the rise time constants, $\phi_m$ is the membrane potential, and $E_{Na}$, $E_K$, and $E_{Ca}$ are the $Na^+$, $K^+$, and $Ca^{2+}$ reversal potentials, respectively. Parameter values are listed in Table 6.

The spike times $t_s$ were given by a Poisson spike train, which we modeled using the `homo-geneous_poisson_process` function from the Elephant software package [118] in Python. The function takes the synaptic input rate, the synaptic starting time, and the synaptic stopping time as arguments. The values of these parameters are given in the labels and figure caption of Fig 7.

Attaching synapses to the neuron, changed the ion dynamics in the following way:

$$\frac{dN_{Na,i}}{dt} \quad \rightarrow \quad \frac{dN_{Na,i}}{dt} - \frac{I_{syn,Na}}{Fz_{Na}}, \tag{136}$$

$$\frac{dN_{Na,e}}{dt} \quad \rightarrow \quad \frac{dN_{Na,e}}{dt} + \frac{I_{syn,Na}}{Fz_{Na}}, \tag{137}$$

$$\frac{dN_{K,i}}{dt} \quad \rightarrow \quad \frac{dN_{K,i}}{dt} - \frac{I_{syn,K}}{Fz_{K}}, \tag{138}$$

**Table 6. Synaptic parameters.**

| Parameter | Value | Reference |
|---|---|---|
| $\overline{g}_{\mathrm{syn,Na}}$ | $1.0 \cdot 10^{-9}$ S | |
| $\overline{g}_{\mathrm{syn,K}}$ | $1.9 \cdot 10^{-9}$ S | † |
| $\overline{g}_{\mathrm{syn,Ca}}$ | $6.5 \cdot 10^{-12}$ S | ‡ |
| $\tau_1$ | $3.0 \cdot 10^{-3}$ s | [117] |
| $\tau_2$ | $1.0 \cdot 10^{-3}$ s | [117] |

† $\overline{g}_{\mathrm{syn,K}}$ was calculated from the criterion that $I_{\mathrm{syn,K}} = \frac{1}{2}|I_{\mathrm{syn,Na}}|$ at steady state ($\phi_\mathrm{m} = -66.9$ mV). The same Na$^+$/K$^+$ current fraction was previously used in an NMDA-synapse model [39].

‡ $\overline{g}_{\mathrm{syn,Ca}}$ was calculated so that $I_{\mathrm{syn,Ca}} = 0.02(I_{\mathrm{syn,Na}} + I_{\mathrm{syn,K}} + I_{\mathrm{syn,Ca}})$ at steady state, i.e., so that a small amount (2%) of the AMPA current was carried by Ca$^{2+}$.

$$\frac{dN_{\mathrm{K,e}}}{dt} \quad \rightarrow \quad \frac{dN_{\mathrm{K,e}}}{dt} + \frac{I_{\mathrm{syn,K}}}{Fz_{\mathrm{K}}}, \tag{139}$$

$$\frac{dN_{\mathrm{Ca,i}}}{dt} \quad \rightarrow \quad \frac{dN_{\mathrm{Ca,i}}}{dt} - \frac{I_{\mathrm{syn,Ca}}}{Fz_{\mathrm{Ca}}}, \tag{140}$$

$$\frac{dN_{\mathrm{Ca,e}}}{dt} \quad \rightarrow \quad \frac{dN_{\mathrm{Ca,e}}}{dt} + \frac{I_{\mathrm{syn,Ca}}}{Fz_{\mathrm{Ca}}}. \tag{141}$$

## Analysis

To calculate $\phi_{\mathrm{se,n}}$, $\phi_{\mathrm{se,g}}$, and $\phi_{\mathrm{se,diff}}$, we used our calculations from the analysis section in [33] as a starting point. In [33], we split $\phi_{\mathrm{se}}$ into two components:

$$\phi_{\mathrm{se}} = \phi_{\mathrm{se,VC}} + \phi_{\mathrm{se,diff}}, \tag{142}$$

where $\phi_{\mathrm{se,VC}}$ is the potential given by standard volume conductor (VC) theory, and $\phi_{\mathrm{se,diff}}$ is the additional contribution arising from diffusive currents. They are calculated from

$$\phi_{\mathrm{se,VC}} = i_\mathrm{e} \frac{\Delta x}{\sigma_\mathrm{e}}, \tag{143}$$

and

$$\phi_{\mathrm{se,diff}} = -i_{\mathrm{e,diff}} \frac{\Delta x}{\sigma_\mathrm{e}}, \tag{144}$$

where $i_\mathrm{e}$ is the extracellular axial current density, and $i_{\mathrm{e,diff}}$ is the diffusive component of the extracellular current density.

In the edNEG model, currents travel in two connected loops: one going in and out of the neuron, and one going in and out of the glial domain (Fig 1). Current continuity then requires that:

$$i_\mathrm{e}A_\mathrm{e} = i_{\mathrm{msn}}A_\mathrm{m} + i_{\mathrm{msg}}A_\mathrm{m}, \tag{145}$$

and

$$i_e A_e = -i_{mdn} A_m - i_{mdg} A_m. \tag{146}$$

If we insert Eq 146 into Eq 143, we get

$$\phi_{se,VC} = -i_{mdn} \frac{A_m}{A_e} \frac{\Delta x}{\sigma_e} - i_{mdg} \frac{A_m}{A_e} \frac{\Delta x}{\sigma_e}, \tag{147}$$

that is, $\phi_{se,VC}$ may be split into two components, $\phi_{se,n}$ and $\phi_{se,g}$, where

$$\phi_{se,n} = -i_{mdn} \frac{A_m}{A_e} \frac{\Delta x}{\sigma_e}, \tag{148}$$

and

$$\phi_{se,g} = -i_{mdg} \frac{A_m}{A_e} \frac{\Delta x}{\sigma_e}. \tag{149}$$

Note that $i_{mdn}$ and $i_{mdg}$ denote the sum of all transmembrane currents in the respective compartments, including the capacitive currents.

The extracellular potential gradients are given by $\Delta\phi_{e,n} = -\phi_{se,n}$, $\Delta\phi_{e,g} = -\phi_{se,g}$, and $\Delta\phi_{e,diff} = -\phi_{se,diff}$. If we take the sum of $\phi_{se,n}$, $\phi_{se,g}$, and $\phi_{se,diff}$, we get the extracellular potential $\phi_{se}$ as calculated using the KNP framework.

## Numerical implementation

We implemented the code in Python 3.6 and solved the differential equations using the `solve_ivp` function from SciPy with its Runge-Kutta method of order 3(2). We set the maximal allowed step size to $10^{-4}$ s in all simulations, except when we simulated physiological activity in Fig 6, where we set it to $10^{-5}$ s. The code can be downloaded from https://github.com/CINPLA/edNEGmodel and https://github.com/CINPLA/edNEGmodel_analysis.

## Supporting information

**S1 Fig. Membrane currents during physiological activity.** Components (various ion channels, stimulus, capacitive currents, and ion pumps) of the transmembrane current in the neural soma layer **(A)**, the neural dendrite layer **(B)**, the glial soma layer **(C)**, and the glial dendrite layer **(D)**. The current components were plotted as moving averages using a time window of 10 s. The simulation was the same as in Fig 3. **(A)** The neuronal membrane current was dominated by the injection stimulus current (sink) in the soma-layer during firing. Among the other currents, the delayed rectifying $K^+$ current contributed the most, but the other (ionic) current components were on the same order of magnitude. As expected, the capacitive current ($\propto d\phi_m/dt$ averaged over 10 s) was close to zero. After firing ceased ($t > 600$ s), the membrane current was dominated by the pump and leak currents, being oppositely directed, keeping the cell in a steady resting state. **(B)** The afterhyperpolarizing $K^+$ current dominated the neuronal membrane current (source) in the dendrite-layer during firing, but the other (ionic) currents were on the same order of magnitude. In the steady resting state, the pump- and leak currents dominated the membrane current, just like in the soma-layer. **(C-D)** In both glial compartments, the (inward) leak current was the largest component, closely, but not entirely, balanced by the (outward) Kir and pump currents. Whether the total membrane current amounted to a glial source (soma-layer) or sink (dendrite-layer)

was determined by the magnitude of the Kir current.
(TIF)

**S2 Fig. Membrane currents during pathological activity.** Components (various ion channels, stimulus, capacitive currents, and ion pumps) of the transmembrane current in the neural soma layer **(A)**, the neural dendrite layer **(B)**, the glial soma layer **(C)**, and the glial dendrite layer **(D)**. The current components were plotted as moving averages using a time window of 10 s. The simulation was the same as in Fig 4, where the neuron was driven into depolarization block. **(A)** At the end (and throughout most) of the simulation, the neuronal membrane current in the soma-layer was primarily composed of an (outward) $Na^+$ current, and the (inward) delayed rectifying $K^+$ current and pump current. The $Na^+$ current was largest in magnitude, but smaller than the sum of the two inward currents, so that the soma was a net current source. **(B)** At the end of the simulation, the neuronal membrane current (sink) in the dendrite layer was dominated by the (inward) $Ca^{2+}$ current. **(C)** The glial membrane currents were dominated by the leak current (sink) in the soma-layer and **(D)** the Kir current (source) in the dendrite-layer.
(TIF)

## Author Contributions

**Conceptualization:** Marte J. Sætra, Geir Halnes.

**Formal analysis:** Marte J. Sætra, Geir Halnes.

**Funding acquisition:** Gaute T. Einevoll.

**Investigation:** Marte J. Sætra, Geir Halnes.

**Methodology:** Marte J. Sætra, Geir Halnes.

**Project administration:** Gaute T. Einevoll, Geir Halnes.

**Software:** Marte J. Sætra.

**Supervision:** Gaute T. Einevoll, Geir Halnes.

**Validation:** Marte J. Sætra.

**Visualization:** Marte J. Sætra, Gaute T. Einevoll, Geir Halnes.

**Writing – original draft:** Marte J. Sætra, Geir Halnes.

**Writing – review & editing:** Marte J. Sætra, Gaute T. Einevoll, Geir Halnes.

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
