## [Decision Letter · Decision Letter 0]

8 Oct 2020

Dear Dr. Halnes,

Thank you very much for submitting your manuscript "An electrodiffusive neuron-extracellular-glia model with somatodendritic interactions" for consideration at PLOS Computational Biology.

As with all papers reviewed by the journal, your manuscript was reviewed by members of the editorial board and by several independent reviewers. In light of the reviews (below this email), we would like to invite the resubmission of a significantly-revised version that takes into account the reviewers' comments.

Although the reviewers acknowledge the interest of the developed model, they express doubts about its novelty, in particular compared to your recent paper Sætra et al, PLoS CB, 2020. They also criticise the weakness of the biological information that its gained from the model. 

We cannot make any decision about publication until we have seen the revised manuscript and your response to the reviewers' comments. Your revised manuscript is also likely to be sent to reviewers for further evaluation.

Sincerely,

Hugues Berry

Associate Editor

PLOS Computational Biology

Daniele Marinazzo

Deputy Editor

PLOS Computational Biology

Reviewer's Responses to Questions

**Comments to the Authors:**

Reviewer #1: The manuscript "An electrodiffusive neuron-extracellular-glia model with somatodendritic interactions” by Satra, Gaute T. Einevoll and Geir Halnes uses numerical simulations of electrodiffusion model to investigate the spatial-temporal communication between neuron and glia cells.

Following the logic and results of the manuscript, I guess that the most important achievement is a model itself. Readers can find a new electrodiffusion model of glia - neuron intercommunication via intra- and extracellular ionic concentrations. The suggested model is realistically representing the complex phenomenon in a space between neuron and glia and, from my point of view, can be the essential step in the definition of existing hidden conversations between cells living in the electrolyte.

The main benefit of the manuscript is the model itself: attractive, well-described, with details mechanisms based on principles of electrodiffusion, including critical parameters of geometrical configurations. The authors highlighted critical limitations of the model also.

Theorists will benefit the manuscript if accepted, will actively cite and use as desk information when someone needs to find the right equation or the correct parameter. The work is accompanied by program code in Python, which adds advantage for this article. Everyone simulating with Neuron will be happy to include some of the scripts to their models.

My opinion is that the model deserves the very highest recommendation. The model is in some way similar to the already published by authors “An electrodiffusive, ion conserving Pinsky-Rinzel model with homeostatic mechanisms Marte J Sætra, Gaute T Einevoll, Geir Halnes”.

The drawback of this project, like many other similar projects, is the triviality of the results. The authors first wrote a new exciting model, and then began to think about what to do with it, and not vice versa. For example, after discovering a new phenomenon that cannot be explained using the old paradigm authors have proposed a new theory (model) that explains the new phenomena.

However, for the sake of fairness, in general, the results may be of interest to a wide range of readers working in the field of electrolyte dynamics, cell biophysics, and electrophysiology.

To improve the manuscript, the authors should explicitly specify a working scientific hypothesis and identify the main scientific results of the project. The paper should have a logical organisation that reflects the aims or research questions of the manuscript, including any hypotheses that have been tested. The model results needs to make connections with experimental data.

A few comments on the text and the artwork

Introduction.

1. Lines 38-39. The statement “Until recently, models that were consistent in this regard (see, e.g., [16, 27{29]), had not 39 accounted for morphological aspects of neurons “ is not correct, please see the paper of J. Cartailler, D. Holcman Electrodiffusion Theory to Map the Voltage Distribution in Dendritic Spines at a Nanometer Scale Neuron, 2019 Nov 6;104(3):440-441. doi: 10.1016/j.neuron.2019.10.025.

2. Fig.1 Authors defines volume fractions as 0.4/0.2/0.4. Since this is a critical parameter of the electrodiffusion model that will able to change results, would be nice to have an explain of this numerical value and the citations here are critical.

3. Fig.2. To compare two models A-C and B-D please plott results at the same time frames?

4. Fig.3. “The glial domain did not contain any Ca2+ channels.” Glia has a lot of Ca channels, why authors excluded the Ca channels from glia?

5. Table 1. I recommend micrometres for dimension. Why geometrical parameters in the table, which in general have no physical measurement, are deduced with such precision, and not just 1000, 500 and so on. Was there any meaning in that or the value for some model calculations, stability, for example.

6. Eq. 118 and 119, is it a typo?

7. The question related to the equations 4-9. I was not able so to found how the authors solve the problem associated with electrodiffusion in conditions of non-uniform geometry, for example, in case of the transition from the soma to the dendrite, the place where the number of ions can be preserved, but not concentrations since a jump of geometry. In this case, membrane potential will also be jumped, which leads to a non-uniform electric lateral current.

8. The Eq.63 and 64. The electrical current of any co-transport, Itr, is determined by product of a membrane conductivity and a deviation of membrane potential from the Nernst potential, Itr=G(V-Enernst). However, in the manuscript, The current of co-transporters (Eq 63 ) is the multiplication of conductivity by the Nernst potential and thus does not depend on the membrane potential, and Eq 64 depends on membrane potential but in a nonlinear way. Please explain.

9. I would recommend rewriting equations 70-73 with the right-hand side explicitly dependenting on a volume.

Reviewer #2: Reproducibility report has been uploaded as an attachment.

Reviewer #3: The paper proposes a framework for modeling of neuronal excitation and ionic dynamics in a macroscopically homogeneous neural tissue that includes neurons, glial cells and extracellular space (ECS). Because of importance of spatial extension of neurons, the neurons are considered as two-compartmental ones surrounded by different subpopulations of glial cells and different compartments of ECS. Such 2-by-3 compartment description seems to be well equilibrated. The authors introduce a convenient set of notations for main variables and formulate the model with canonical equations of electrical circuits and ionic channels, as well as with a phenomenological approximation of volume change. Such formulation pretends to be a standardized one, and seems to be useful for programming.

Concerns:

- I agree that the distinction of somatic and dendritic compartments is critically important. The reasons (not mentioned in the manuscript) are: (i) comparison and reproduction of experimental patch-clamp recordings in voltage- and current-clamp modes require at least 2-comp. model; (ii) calculation of local field potentials requires at least 2-comp. model; (iii) ionic dynamics is different in somatic and dendritic compartments of neurons, because of inhomogeneous distribution of synaptic receptors. I also agree that the glial cells adjacent to different neuronal compartments may show different dynamics and are probably worth to be distinguished. Thus, the proposed 2x3-model seems to be reasonable. However, synaptic interaction between neurons is much more powerful in sense of contribution to both the neuronal excitation and the ionic dynamics. Unfortunately, the synapses are absolutely ignored in the paper, which devaluates the work.

- In its present form, the model seems to be just a slightly updated version of the very recently published model of the same authors [17]. The key moment of their model description is an integral form of the equations for voltages. Including glia is straightforward and known, in particular, from their previous work [38]. Its effect has been shown before. A novelty of the present paper is two compartments of glia, but the authors do not refer to those effects of soma- and dendrite-adjacent glial compartments that could be revealed in experiments.

- The model does not take into account the lateral diffusion of ions and their diffusion in the bath solution. The latter is quite necessary for comparison with slice experiments, however, it may destroy the conservative form of the equations.

- The initial results are focused on Ca-spikes, whereas the later results are on the model of spreading depression (SD) in the absence of synaptic activity, i.e., presumably, the case of experiments with zero-Ca solution. I see a contradiction between these two targets.

- Stimulation of the neuron with only potassium current (Eqs.118 and 119), as if with a pure potassium pump, seems to be unrealistic.

- The authors consider as an advantage of the model that it reproduces spike shapes obtained in rather old Pinsky&Rinzel’s modeling work. To my opinion, those shapes are not common, if searching within recent patch-clamp recordings. Instead, it would be valuable to validate the simulated effects of ionic concentration shifts on neuronal activity with recent experiments.

- Am I right that the low firing frequency (about 1Hz) of the neuron is due to the AHP-channel (beta_q=1)? If so, it seems to be unrealistic. A plot of spiking frequency versus current and shunting conductance (or current at least) would be helpful to characterize the neuron.

- The model considers ``a local piece of neuro-glial brain tissue”, and any considerable shift of ionic concentrations in ECS is expected to originate not from a single neuron but from many neurons in some vicinity. However, as written, for instance, in the ``Homeostatic breakdown in the edNEG model” section, only a single neuron is stimulated: ``There, the neuron received a strong input current…”. I guess, the authors mean the stimulation of all neurons simultaneously, which relates to Figs.3-6.

- It is written concerning the depolarization block that ``This kind of dynamics can not be captured with standard neuron models constructed under the assumption that ion concentrations remain constant under the simulated period.” It is not true.

- The authors do not observe recovery from depolarization block. They suggest that ``the recovery might be observed if the edNEG model were expanded to a spatially continuous model”. I guess, it is sufficient to take into account the potassium relaxation to its concentration in the bath solution, as done in many previous models, like in [44] and [68].

- Synaptic currents provide important contributions to SD, at least NMDAR-mediated currents (Somjen 2001). These currents provide one of the major sources of sodium and potassium ionic fluxes. Therefore, it is not correct to simulate the ionic dynamics during SD without taking NMDA-currents into account. (Though, it has been done in some previous works.) In its present version the model might pretend to reproduce some slice recordings under the conditions of synaptic receptor blockage and equal for all neurons excitation, presumably, provided with optogenetics.

- Panels A and C of Fig.6 demonstrate the effect of glia. However, if the potassium concentration in glia changes only because of K-IR channels and the pump, then in a stationary regime the total potassium flux into glia is zero, and thus the glia should not affect the potassium concentration in ECS. Neglecting by the volume change, I conclude that the glia just occupies some volume thus reducing the ECS volume. So, the effect of glia seems to be trivial, it is just an effect of a single parameter, which is the ratio of neurons’ to ECS’s volumes.

- The authors explain AHP observed in Fig.6B by the effect of the pump and compare with the experiment from Gulledge et al. 2013 [49]. Though the magnitude of the effect seems to be similar (a few mV), it is not the case. The AHP in [49] takes place in the resting, low-conductance state, whereas similar voltage shift in Fig.6B is observed in the regime of high conductance. This fact points to overestimation of the pump density in the model.

- Spreading depression is a spreading phenomenon, but it is modeled as a localized event, which, however, is a common place of models.

- The volume dynamics is driven by the osmotic pressure. At the same time, it is known that neurons do not have aquaporins (Murphy // Front. Cell. Neurosc. 2017) and react to osmotic pressure change weaker than to the change of potassium concentration [Andrew et al. // Cer.Cortex 2007]. The choice of the osmose-based model for the volume dynamics in neurons is not justified.

- The model neglects the flux of water between somatic and dendritic compartments of both neurons and ECS. Why? The reference to [95], where the macroscopic flow is considered, seems to be irrelevant here.

- The constructed model is aimed to provide ``a general framework for combining multicompartmental neural modeling with electrodiffusive ion concentration dynamics in neuroglial brain tissue”, whereas ``many previous models have parts of the same functionality”. Below I have listed some most representative works of different teams of authors and compared the details included and not included in their models.

[79] – Krishnan et al. 2015 – net, 2 comp. + glia + KCC2 + propagation in 1D + synapses, but not syn. contribution in the ionic dynamics, no transporters

[68] Wei et al. 2014 “Oxygen…” – network, + synapses +volume, but not syn. contribution in the ionic dynamics

[44] Wei et al. 2014 “Unification…” – 1 comp. + glia + KCC2, NKCC1 + volume + O2

Gentiletti and Suffczynski // IJNS 2017 – E and I, 3 comp. + glia, but no synapses

Chizhov et al. // Plos One 2019 – network, 2 comp. + glia + NKCC1, KCC2 + synapses contributing to ion dynamics, but no spatial propagation, no volume

[13] Kager et al. 2000 – distributed neuron + buffer + K-dependent NMDA

[60] Florence et al. 2009 - 1 comp. + glia, but no synapses, no transporters, no spatial propagation

[75] Chang et al. 2013 – 2-comp. + glia + O2 + blood + propagation in 1D, but no synapses, no transporters

[27] Tuttle et al. 2019 – 1-comp. + glia + propagation in 2D + NMDA contributing to ion dynamics + volume, but no network

Unfortunately, in its present version the constructed model also has only parts of the target functionality.

Minor comments:

- Abbreviation KNP appears in Results before its introduction in Methods.

- Black square in Fig. 3A looks weird.

- One of the labels ``I” in the caption to Fig.6 should be ``J”.

- Formulas relating a concentration to an amount of substance and a volume is given only in words (after eq.10), which is inconvenient for readers.

- “Constraint number (iv)” should be `` Constraint number (4)” before eq.17.

- Hodgkin-Huxley approximations are written as dependent everywhere on the voltage phi_m. Should it be phi_{dn} for the dendritic currents and phi_{sn}-phi_{se} for the somatic currents?

- Similar, the volume in eq.66; the voltage and concentrations in eqs.67-69.

- M is not given in eq.76.

Reviewer #4: The authors present a computational model of a (layered) two-compartment neuron (soma-dendrite) and (two) glia cells connected both to a two compartmental extracellular space. The neuron and the glia cell have a standard collection of ionic conductances and ionic pumps/exchangers that allow not only to calculate the dynamics of electrical currents and membrane voltage, but also all ionic concentrations and fluxes between the compartments. They demonstrate in their results that this model is “stable” at a defined resting state, the neuron can fire and goes into depolarization block already at a relatively low firing rate of ~8 Hz. They also incorporated the consequences of changing osmotic pressure as a consequence of ion changes in the compartments: swelling and shrinking of those compartments under “normal” physiological conditions and under the extreme pathological condition of spreading depression.

The authors present a solid piece of work in a well-written and well-documented paper. I think the model they present is not as new as they claim (almost any aspect of it has already been presented in published work by others); however this model has the potential to add substantially to our knowledge of the functional consequences of neuron-glia interactions and the role of ion(concentration)s in the nervous system. One of the properties in which this model can surpass the already published (NEURON environment) models is that it is capable of calculating the extracellular potential(gradient)s. Unfortunately, the authors only mention this in a hidden sentence and do not expand it into something useful (see below, under compartments). The main body of the paper describes the architecture of the model, composed of the combination of two well-validated models of a neuron and a glia cell. The validation of the complete model in the results is not very strong and more an illustration than a test against known experimental facts. More than half of the discussion is spent on presenting potential new applications of the model in analyzing spreading depression. If this part of the paper would be moved to the results and seriously designed and investigated, it would considerably enhance the impact of this paper/model.

The proposal in the discussion to use the tri-domain-model as a basic element to create a larger system (and also investigate the spreading in “spreading depression”) is suggestive, but also a huge oversimplification of what needs to be done. The model, however, is potentially better capable of investigating ephaptic interactions between neurons than many of the existing ones. For the currently presented single neuron situation the extracellular potential gradients are too small to be relevant, however we know that in regions like CA1, CA3 and DG in the hippocampus we encounter extracellular field potentials of several millivolts, well capable of affecting the phenomena under study here.

A similar remark holds for the glia cell. The way the coupling is now proposed in the tri-times two domain model in the discussion does not pay tribute to the power of spatial glia buffering. The current model is implemented as a “closed system”, but we know that for the phenomenon under study (spreading depression) K+ is virtually transported to the EC outside the realm of the neuron by the glia syncytium: very strongly electrically coupled groups of glia cells. In a more primitive implementation K+ can be temporarily buffered in the blood stream; a mechanism that is certainly relevant for the spreading of SD and again a realm where this model could fill an interesting spot.

My strong suggestion is to at least incorporate one of these aspects in the result section.

I also suggest that the authors are more careful in calling stabilizing mechanism homeostatic: I prefer to reserve the latter word for mechanisms that at least seem to have some kind of “set-point” that they try to maintain. Most mechanism that are mentioned in this paper under homeostatic have indeed a stabilizing effect, but to the best of my knowledge they do not try to preserve a defined set-point. In fact the authors need to tweak the “leak currents” in order to attain a stable resting membrane voltage. This is often done in this type of modelling and I do not object. However critical questions about the physiological meaning of these ion-selective non voltage-dependent channels cannot be answered. Calling them leak is to a large extend hiding the problem!

The adequacy of a model and its correctness can never be judged without taking into consideration its detailed purpose. As far as I can judge this model is supposed to present a realistic firing pattern for a CA3/CA1 hippocampal neuron and should contain the elements necessary to describe the depression aspect of the phenomenon of “spreading depression”. The authors suggest that the model can be universally applied to solve many questions. However, for that aim at least a few elements need to be added (and probably a motivation why they are not incorporate would clear things up and warn potential users!).

-- The transient A-current and the hyperpolarization activated h-current are two potassium currents with a very important functional role in the firing patterns of these pyramidal neurons.

-- It is unclear why there is no Na current in the dendrites: this prevents any realistic study on phenomena where back-propagating APs are relevant.

-- The lack of any Ca current in the soma (maybe even the low and the high threshold variants) and an IC-AHP K current in the soma might be the reason that we see hardly any of the normally observed frequency adaptation in these neurons. In my opinion these currents need to be present if one wants to undertake a study of depolarization block. All pyramidal cells that I have recorded from easily produced firing rates in the 10-20 Hz range without ever going into depolarization block.

-- The implementation of the model implies a reversal potential for chloride in rest of -90.3 mV. That is extremely low and not supported by experimental evidence. As there are no chloride channels it also assumes a very high chloride pumping rate in the rest situation.

-- It would be helpful for the experimentalists if the authors reported the input-impedance of their neuron and the glia cell in the resting state.

Whether or not the above-mentioned omissions/additions are relevant, strongly depend on the phenomena one wants to describe and the precision one wants to attain. More serious are the points mentioned below as they deal with fundamental errors in the model:

-- As is done very often in modeling the authors use the Nernst equation to calculate the reversal potential and then Ohm’s law to describe the ionic current. This is only correct in the equilibrium situation. Currents here should be calculated using the Goldman-Hodgkin-Katz equation. I do not think that this will strongly affect the conclusions, but it will considerably change the numbers. If that is not relevant, why then do we try to make correct models?

-- The authors use an (artificial) K current injection into the soma to depolarize the neuron and bring it to firing. In real life firing is mostly brought about by synaptic activation, implying a substantial change in synaptic conductance with current flow and depolarization as a consequence. This looks like a minor difference, but the change in conductance affects the membrane time constant and thus the dynamics of the currents. It also implies that most activity starts from the dendritic compartment (although during SD, the shifts in ionic composition will directly initiate somatic firing). In the current model synaptic conductances are not implemented, but there is hardly a phenomenon where they are not involved….

-- The authors assume the same volume fractions for the somatic and the dendritic compartments, which is clearly not the case and experimentally well documented. It might explain why they notice little difference in their soma versus dendritic observations (fig 5).

-- A similar remark holds for the substantial difference that should exist between the somatic surface-volume ratio, compared to the dendritic surface-volume ratio. There are a few vague remarks on this point, but the consequence should be a considerable difference in current-inflow to concentration change in the two regions (probably hidden because the lack of Ca2+ influx in the soma and synaptic ionic influx in the dendrites).

-- The authors claim that two compartments in the neuron catch the essentials of the neuron’s behavior, but also that adding more compartments is not a big deal. The latter improvement is needed if we want a better inside into the extracellular potential gradients and if we want to implement IA and Ih which have a specific gradient distribution in the dendrites of these cells. Such expansions are relatively easy done in modeling environments as NEURON, but a lot harder in Python code!

-- The authors attain conservation of particles in their model but have a huge problem to attain electroneutrality in all compartments. They solve it by adding impermeable anions which is OK. However they need 97% of the anions in the neuron to be impermeant and 95% in the glia, which is far more than experiments suggest. They also add 16 mM impermeant anions to the extracellular space, for which there is no experimental support whatsoever, to the best of my knowledge…

-- I understand that the purpose of the illustrations in the results is more to illustrate than to validate against experimental data, but is about 0.5 mM increase in extracellular K+ for each AP of a single neuron not a bit much? (Fig 3B).

**Have all data underlying the figures and results presented in the manuscript been provided?**

Reviewer #1: Yes

Reviewer #2: None

Reviewer #3: Yes

Reviewer #4: Yes

PLOS authors have the option to publish the peer review history of their article (what does this mean?). If published, this will include your full peer review and any attached files.

Reviewer #1: No

Reviewer #2: **Yes: **Anand K. Rampadarath

Reviewer #3: No

Reviewer #4: No
---

## [Decision Letter · Decision Letter 1]

27 Apr 2021

Dear PhD Halnes,

Thank you very much for submitting your manuscript "An electrodiffusive neuron-extracellular-glia model for exploring the genesis of slow potentials in the brain" for consideration at PLOS Computational Biology. As with all papers reviewed by the journal, your manuscript was reviewed by members of the editorial board and by several independent reviewers. The reviewers appreciated the attention to an important topic. Based on the reviews, we are likely to accept this manuscript for publication, providing that you modify the manuscript according to the review recommendations.

Sincerely,

Hugues Berry

Associate Editor

PLOS Computational Biology

Daniele Marinazzo

Deputy Editor

PLOS Computational Biology

[LINK]

Reviewer's Responses to Questions

**Comments to the Authors:**

Reviewer #1: The manuscript "An electrodiffusive neuron-extracellular-glia model with somatodendritic interactions” by Satra, Gaute T. Einevoll and Geir Halnes has been improved, and now it is easier to follow the logic and ideas of the manuscript. I think that the result is of interest for experts in the area that deals with simulation of astrocyte-neuron interactions.

I have no technical objections to the publication of the results.

I gratitude the authors for access to the programming code via an online repository Github, to better understand the model.

Reviewer #2: The Reproducibility Report has been uploaded as an attachment.

Reviewer #3: The review is uploaded as an attachment

Reviewer #4: An electrodiffusive neuron-extracellular-glia model for exploring the genesis of slow

potentials in the brain

Saetra et al.

PCOMPBIOL-D-20-01192R1

This is probably one of the best revisions that I have seen in my career as a reviewer. I particularly want to thank the authors for their extensive and well thought answers on all my questions, even those that are no longer relevant for the present version of the manuscript.

One of the main criticisms of the previous version of this paper was that it was to much a model for the model paper, which will restrict the audience to only a relatively small group of actual modelers. The authors have now made a revision, which still presents the same interesting model, but they have focused much more on real neuroscience questions then on speculations of for what the model could be used for. The main issue is now on (small) slow potentials in nerve tissue (and a bit on depolarization block).

As any good paper raises more questions than it can answer, a have a few points to make. Except for one question out of interest, they do not demand additional experimental work, and most can probably be easily handled at a textual level.

-- In the description and the potential use of the model (abstract and introduction/discussion lines 459-477) it might be helpful for potential users to explicitly state that the model is restricted to polarized neurons organized in a layered palisade way: typically, hippocampus regions. Five-layered cortex is not impossible but would be quite a programming challenge!

-- At several points in the paper the authors generate and discuss depolarization block. I fully agree with those points, but it should be mentioned that the model is not optimized to handle that topic. In my experimental experience it is rather hard to induce depolarization block in hippocampal pyramidal neurons. The model does not contain somatic calcium channels coupled to strong potassium channels that produce slow and fast after hyperpolarization and spike frequency adaptation. They limit firing rate and prevent depolarization block to a large extend. It is not a main point of the paper but should be discussed.

-- The authors induce activity in the neuron either by current injection or by synaptic activation. Current injection is done by transferring K+ (or Na+/Cl-) ions from the extra- to the intra-cellular compartment. This guarantees conservation of ions but invalidates electroneutrality in both compartments. Is that physically realistic and acceptable? Please comment.

Stimulation is always followed by re-equilibration of the ions and we noticed in our own (unpublished) models that due to the small Cl- currents this can be a very slow process.

I appreciate the way synaptic input is implemented, as it avoids the problem mentioned above. There is, however, also here one peculiarity and that relates to the Mg2+ dependence of the NMDA synaps. Mg2+ is not present in the model so the authors set it to a fixed level of 1 mM. As it is membrane impermeable, it will undergo huge changes in concentration with the volume changes of the EC compartment. In reality [Mg] is often high enough to saturate the binding; it does not seem to be implemented that way here. It could also be that in the current studies (except the depolarization block) the voltage never reaches values sufficient to release the block. Using a glutamate synaps would have avoided this inconsistency and lead to the same conclusions.

-- In lines 407-417 the authors compare the depolarization block state with a Donan equilibrium. To the best of my knowledge the latter one comes about because of the non-permeable (an)ions inside the cell, after concentration gradients have faded. But the resulting membrane potentials can never produce extracellular gradients. That must be brought about by the ion pumps that are still working. It would be a simple and interesting run to stop the pumps and observe the resulting Donan potentials in all compartments.

-- The (preliminary)results on the slow potentials and their contributors are also quite interesting and promising. Could the authors also indicate what the relative contribution of the different pumping components is?

**Have the authors made all data and (if applicable) computational code underlying the findings in their manuscript fully available?**

Reviewer #1: Yes

Reviewer #2: Yes

Reviewer #3: None

Reviewer #4: Yes

PLOS authors have the option to publish the peer review history of their article (what does this mean?). If published, this will include your full peer review and any attached files.

Reviewer #1: No

Reviewer #2: **Yes: **Anand K. Rampadarath

Reviewer #3: No

Reviewer #4: No

Figure Files:

Data Requirements:

Reproducibility:

References:

---

## [Decision Letter · Decision Letter 2]

28 Jun 2021

Dear PhD Halnes,

We are pleased to inform you that your manuscript 'An electrodiffusive neuron-extracellular-glia model for exploring the genesis of slow potentials in the brain' has been provisionally accepted for publication in PLOS Computational Biology.

Best regards,

Hugues Berry

Associate Editor

PLOS Computational Biology

Daniele Marinazzo

Deputy Editor

PLOS Computational Biology

Reviewer's Responses to Questions

**Comments to the Authors:**

Reviewer #3: I thank the authors for thorough elaboration of the responses to my commentaries. I have no further requests, however let me comment the issue concerning the formula C*V=Q. I admit that from the point of view of mathematics, this formula is just an integral of the differential equations, the Kirchhoff’s law and the ion balance equations, with certain initial conditions. The initial concentrations of ions and the concentration of impermeable anions set the total charge Q to be correspondent to some realistic value of V. So, after this adjustment the formula C*V=Q provides correct calculation of V. Still, from the point of view of physics, where errors of charge estimation are unavoidable, the calculation of V via Q is inapplicable, which makes the formula C*V=Q looking weird. This formula is also misleading, because it does not reflect the true determinants of the resting membrane potential, the leak and the pump rather than the integrals of these terms over time.

**Have the authors made all data and (if applicable) computational code underlying the findings in their manuscript fully available?**

Reviewer #3: None

PLOS authors have the option to publish the peer review history of their article (what does this mean?). If published, this will include your full peer review and any attached files.

Reviewer #3: No

---

## [Editor Report · Acceptance letter]

13 Jul 2021

PCOMPBIOL-D-20-01192R2 

An electrodiffusive neuron-extracellular-glia model for exploring the genesis of slow potentials in the brain

Dear Dr Halnes,

I am pleased to inform you that your manuscript has been formally accepted for publication in PLOS Computational Biology. Your manuscript is now with our production department and you will be notified of the publication date in due course.

With kind regards,

Andrea Szabo
